# Neuronal activation sequences in lateral prefrontal cortex encode visuospatial working memory during virtual navigation

Alexandra Busch [1,2,3,10], Megan Roussy[1,2,4,10], Rogelio Luna [1,2,4], Matthew L. Leavitt[5], Maryam H. Mofrad[2,3], Roberto A. Gulli [6], Benjamin Corrigan [1,2,4], Ján Mináč[3], Adam J. Sachs[7], Lena Palaniyappan [1,8,9], Lyle Muller [1,2,3,11] ✉ & Julio C. Martinez-Trujillo [1,2,4,9,11] ✉

Working memory (WM) is the ability to maintain and manipulate information 'in mind'. The neural codes underlying WM have been a matter of debate. We simultaneously recorded the activity of hundreds of neurons in the lateral prefrontal cortex of male macaque monkeys during a visuospatial WM task that required navigation in a virtual 3D environment. Here, we demonstrate distinct neuronal activation sequences (NASs) that encode remembered target locations in the virtual environment. This NAS code outperformed the persistent firing code for remembered locations during the virtual reality task, but not during a classical WM task using stationary stimuli and constraining eye movements. Finally, blocking NMDA receptors using low doses of ketamine deteriorated the NAS code and behavioral performance selectively during the WM task. These results reveal the versatility and adaptability of neural codes supporting working memory function in the primate lateral prefrontal cortex.

Working memory (WM) is the ability to briefly maintain and manipulate information 'in mind' to achieve a current goal[1]. Brain mechanisms supporting WM represent information in the absence of sensory inputs and without necessarily triggering motor behaviors (see ref. 2 for review). WM differs from long-term memory in that the information is maintained for a short time without necessarily undergoing permanent storage. Lesion and electrophysiological studies in monkeys have implicated the lateral prefrontal cortex (LPFC), a brain area that appears de novo in primate evolution[3], in WM function[2,4]. A long-supported mechanism for coding of visual WM in LPFC is persistent firing in single neurons selective for the memorized information[5,6]. Persistent firing appears sufficient to explain how the brain remembers information about the features or locations of stationary objects for a few seconds[2,4]. However, in settings where memoranda have

spatiotemporal structure, persistent firing may not be sufficient to encode WM[7]. In these cases, LFPC neurons may exhibit alternative coding strategies. This becomes particularly important in naturalistic tasks, where visual scenes are complex and dynamic due to unconstrained gaze.

Interestingly, some studies using delayed response tasks with increased spatiotemporal complexity have reported very few single neurons with persistent firing during the time animals hold a representation in working memory. Instead, many neurons fire transiently, during brief time intervals[8–10]. Some studies have proposed alternative mechanisms to persistent firing, such as short-term synaptic storage[11,12], or oscillatory dynamics[9]. Evidence in favor of such mechanisms remains scarce[13] and it is unclear whether they can encode time varying working memory information. Here, we reasoned

[1]Robarts Research Institute, University of Western Ontario, London, ON, Canada. [2]Brain and Mind Institute, University of Western Ontario, London, ON, Canada. [3]Department of Mathematics, University of Western Ontario, London, ON, Canada. [4]Department of Physiology and Pharmacology, University of Western Ontario, London, ON, Canada. [5]MosaicML, San Francisco, CA, USA. [6]Zuckerman Mind Brain Behavior Institute, Columbia University, New York, NY, USA. [7]The Ottawa Hospital, University of Ottawa, Ottawa, ON, Canada. [8]Department of Psychiatry, University of Western Ontario, London, ON, Canada. [9]Lawson Health Research Institute, London, ON, Canada. [10]These authors contributed equally: Alexandra Busch, Megan Roussy. [11]These authors jointly supervised this work: Lyle Muller, Julio C. Martinez-Trujillo. ✉e-mail: lmuller2@uwo.ca; julio.martinez@robarts.ca

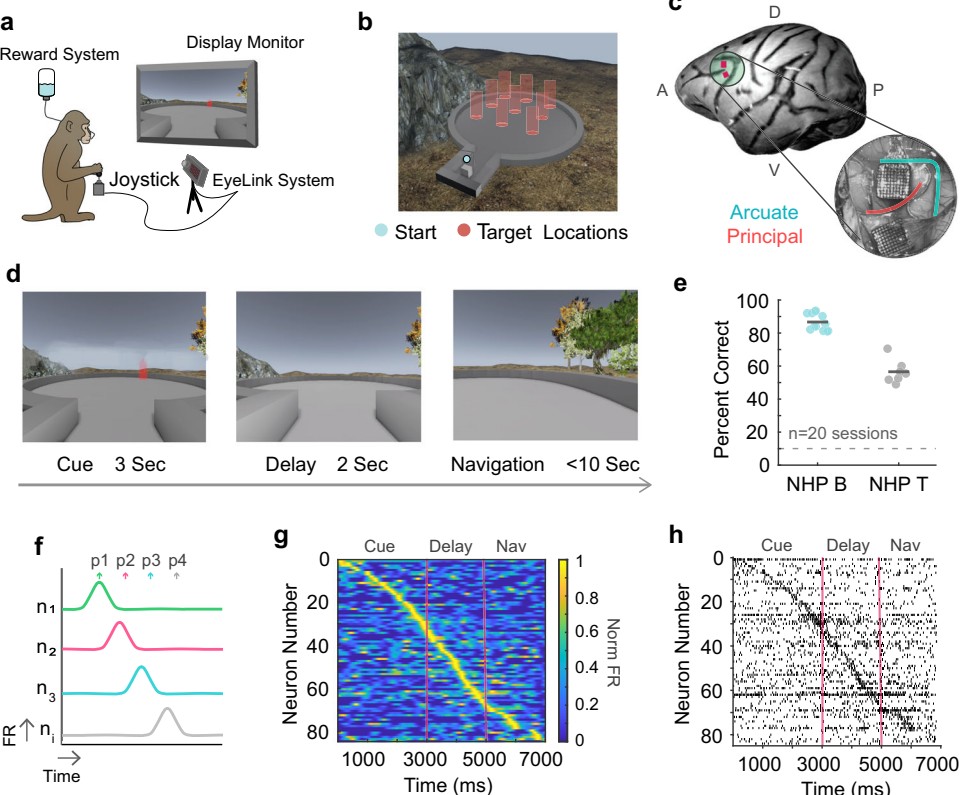

**Fig. 1 | Experimental design for exploration of sequential activity. a** An animal depicted in the virtual reality experimental setup. **b** Overhead view of the nine target locations in the virtual environment. **c** Locational representation and surgical image of the two Utah arrays implanted in the left LPFC of NHP T. **d** Working memory trial timeline. **e** Percent of correct trials for NHP B and NHP T. The dark gray lines represent mean values per animal and the gray dashed line represents chance behavioral performance. Data points represent data from individual sessions. **f** Illustration of temporally tiled activation of individual neurons which may generate sequential patterns of activity at the population level. **g** Normalized firing rates for simultaneously recorded neurons over trial time in one trial. Red vertical lines indicate trial epoch boundaries. **h** Raster plot for the same example trial as 'g' in which each tick represents an action potential.

that a specific mechanism for encoding WM content during spatio-temporally complex tasks may exist. Such a mechanism must be able to encode time-varying information and be robust to interference by sensory and motor signals, such as changes in retinal inputs due to eye movements that happen while maintaining information in WM during naturalistic behaviors.

Neuronal activation sequences (NAS), consisting of temporally precise patterns of single neuron activations, have been reported to encode the varying spatiotemporal structure of motor signals in the high vocal center (HVC) of songbirds[14–18], and spatial trajectories to remembered locations during navigation in the parietal cortex[19] and the hippocampus[20–22] of rodents. Early investigations in macaque monkeys suggested that the spiking activity of a few single neurons in LPFC could have a precise and informative spatiotemporal structure[23]. However, NASs have neither been directly observed in primate LFPC, nor have they been causally linked to WM[13].

Here, we hypothesize that NASs in the LPFC encode specific WM content during spatiotemporally complex tasks. To test this hypothesis, we used microelectrode arrays to record the activity of hundreds of neurons in LPFC layers 2/3 of two macaque monkeys during a naturalistic spatial WM task set in a 3D virtual environment. We focused on the LPFC because of its involvement in WM function[6] and because it has been suggested that coding of WM episodes may involve the prefrontal cortex[24]. During task trials, animals perceived the location of a transient visual cue, remembered that location for a few seconds, and finally navigated toward it to collect a reward using a joystick. We found time boundary neurons that transiently activated just before the beginning and end of the memory period. During the WM period, we found temporally precise NASs that are linked with WM item encoding specifically in the VR task. The link to behavior becomes even stronger when we account for the subject's specific visual perspective within the VR task. We introduce a new and simple decoding algorithm and find that individual WM items can be decoded from the NASs. In the VR task, the NAS encoding outperformed a persistent code based on integrated firing rates of target-selective cells during the delay. In addition, NASs display no link to behavior in an oculomotor delayed response task (ODR) that lacks spatiotemporal complexity. Finally, pharmacological blockade of NMDA receptors with sub-anesthetic doses of ketamine disrupted NASs and selectively impaired WM decoding and task performance.

## Results

We trained two rhesus macaque monkeys on a visuospatial working memory task that took place in a virtual circular arena containing naturalistic elements (see Fig. 1a, b). We recorded neuronal activity using two 96-channel microelectrode Utah Arrays (Blackrock Neurotech, UT, USA) implanted in the left LPFC of both animals (Brodmann area 8a, 9/46[25]) (see Fig. 1c). The task began with a three second presentation of a cue in one of nine possible locations in the arena (cue epoch). The cue then disappeared, and after a two second memory delay period, the animal was required to navigate towards the cued target location using a joystick (see Fig. 1d). Virtual navigation within the environment was exclusively available during the navigation epoch. Animals were able to successfully perform the task (average correct trial rates across sessions were: NHP B: *mean* = 87%, NHP T: *mean* = 57%; chance = ~11%) (Fig. 1e; Supplementary Fig. S1). Eye

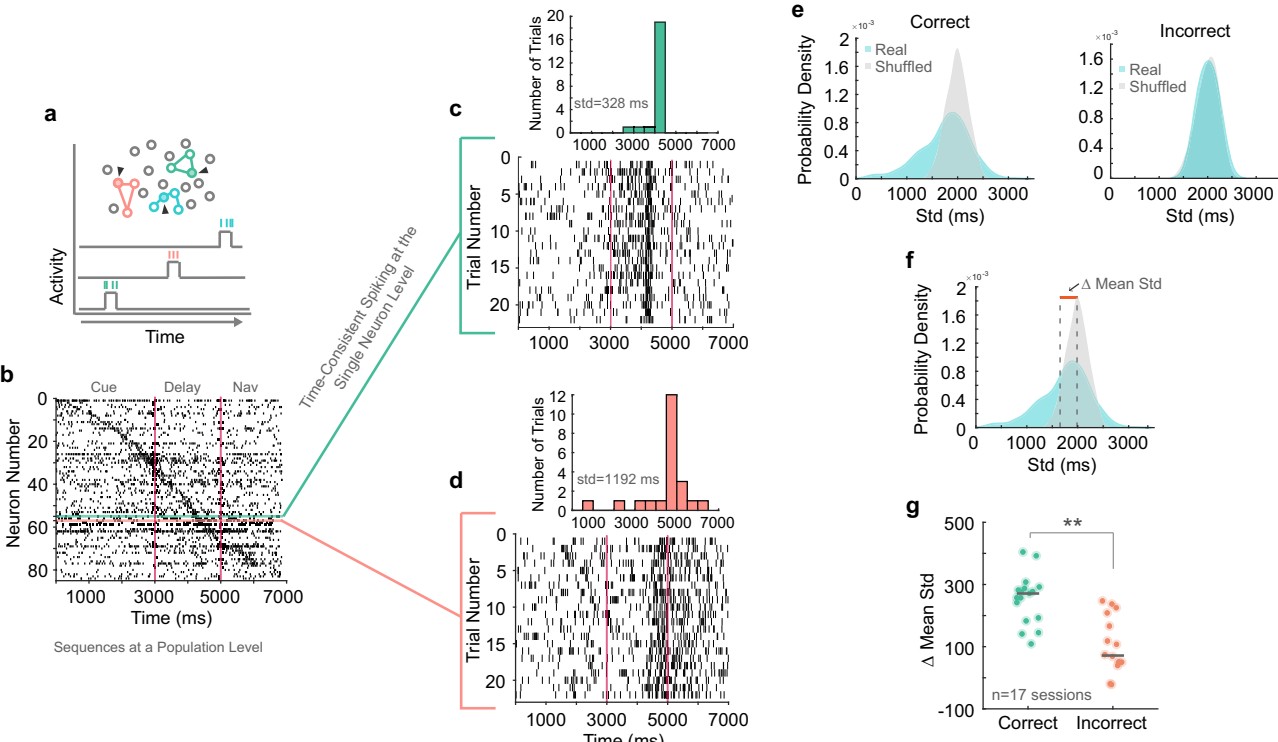

**Fig. 2 | Time consistent neurons underlie sequence formation. a** Depiction of neural ensembles that are activated at different time points throughout a trial. Activity of a single neuron within the ensemble is represented by a bump in activity at a precise time point. Black arrows represent recording electrodes. **b** Single trial raster showing the activity of all simultaneously recorded neurons. Red vertical lines indicate trial epoch boundaries. The green and pink horizontal lines highlight two example neurons. **c** Example neuron one. The raster plot displays action potentials over trial time for this neuron over all trials in a certain target condition. The inlet histogram shows the number of trials in which the max firing time falls within a certain trial time. **d** Represents the same information as 'c' for a second example neuron. **e** Real and shuffled distributions of correct and incorrect trial-trial standard deviations of max firing time for all neurons in an example session. **f** Real and shuffled distributions of correct trial-trial standard deviations of max firing time. Dashed gray lines represent distribution means and the orange line indicates the difference in distribution means. **g** Difference in real and shuffled distribution means for correct and incorrect trials. Each dot represents data from a different session, and dark gray lines represent median values per group. ($p = 0.001$, Wilcoxon signed rank test) *$p < 0.05$, **$p < 0.01$, ***$p < 0.001$.

movements were recorded throughout the task using a video eye tracker. Animals made frequent saccades to explore different scene elements throughout all trial epochs. Firing rates across the recorded neuronal population were poorly tuned for the direction and amplitude of saccades[10,26] (see Supplementary Fig. S2 for eye behavior analyses)

## NASs in LPFC neuronal ensembles

Precise temporal patterns of neural activity have been reported as a mechanism for representing time-varying information in mammalian brains[27]. However, such patterns have not been identified during visuospatial working memory tasks in primates. NASs are typically described as temporally precise activation of neurons above their background rates of activity (Fig. 1f). We observed that LPFC neurons exhibited brief (duration of 80% of max firing value ~ 220 ms) elevations of spike rate above their background levels of firing (Supplementary Fig. S3a) at specific times during the task. We hypothesize that working memory representations during our virtual reality task are encoded by NASs. Specifically, we hypothesize that the timing of these elevated firing events alone could encode WM content, separately from firing rates. To identify potentially relevant population-level patterns in these elevations of spike rate, we sorted neurons by their normalized peak firing time. Sequential patterns emerge in single trials, visualized here using spike density functions (Fig. 1g) and population rasters (Fig. 1h).

A code that relies on NASs implies temporally precise activation of single neurons (e.g., activation of different colored neurons in Fig. 2a

schematic)[27,28]. We examined the firing properties of 3543 neurons in 17 recording sessions (*mean* of 208, *median* of 229 simultaneously recorded neurons per session). Many neurons transiently fired during the same time in correct single trials of the same target condition (Fig. 2b–d, more examples in Supplementary Fig. S3e–j). To quantify this regularity, we calculated the standard deviation (time consistency) of peak firing time between trials of the same condition for each neuron (Fig. 2e, Supplementary Fig. S3b). On correct trials, 20% of neurons (699 neurons) demonstrated a standard deviation below 1000 ms and 65% (2297 neurons) demonstrated a standard deviation below 1500 ms.

We additionally shuffled the peak firing time of each neuron within each trial to generate random firing time estimates across trials. The distributions of standard deviations for correct trials were shifted to lower values relative to the corresponding shuffled distributions (example session in Fig. 2e, all neurons in Supplementary Fig. S3b). The area of non-overlap between the lower tails of the real and shuffled distributions represents the neurons with peak firing times occurring more regularly than expected by chance within trials of the same condition. On the other hand, the real and shuffled distributions overlapped considerably for incorrect trials (example session in Fig. 2e; all neurons in Supplementary Fig. S3b), indicating that neurons' peak firing during single trials of the same condition occurred at less consistent times when animals made mistakes.

Pooling across sessions, we found that the difference between means of the real and shuffled distributions (Fig. 2f) was lower for incorrect trials than correct trials (correct: *median* = 270.9 ms,

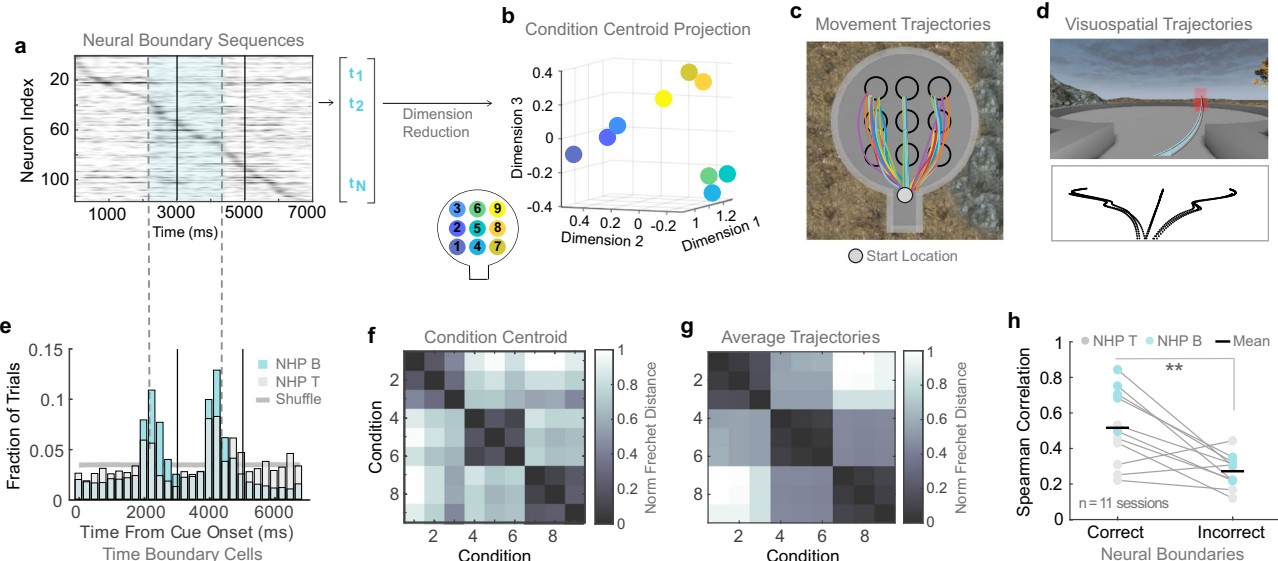

**Fig. 3 | Sequences represent visuospatial trajectories to target locations represent targets and features of subjective visual experience. a** Single trial sequences are represented as spike-time vectors by recording the time within the trial at which each cell reaches its maximum spike density. Solid vertical lines represent the task boundaries dividing the cue, memory delay and navigation epochs. Dashed vertical lines represent the anticipatory neural boundaries dividing these epochs (as determined in 'e'). Only contributions within the neural boundaries (blue) are considered part of the memory period sequence. **b** Dimensionality reduction produces a projection summarizing the trials in each recording session. Here, centroids of same-target trial groups from one example recording session are plotted alongside the layout of the corresponding targets in the virtual arena. **c** Example movement trajectories to targets in the virtual arena, plotted from a

bird's eye view. **d** The virtual arena as seen from the subject's perspective during the delay period, along with average trajectories to each of the 9 targets in this perspective. **e** Firing times of time-selective cells. The two peaks define the neural boundaries between epochs. These peaks correspond to the times at which time-selective cells contributed most to single-trial sequences. **f** Matrix of normalized Euclidean distances between condition centroids in the projection plotted in '**b**'. **g** Matrix of normalized Frechet distances between the average visuospatial trajectories in VR (during the same example recording session as '**b**' and '**f**'). **h** The correlation between the two distance matrices ('**f**', '**g**') for correct compared to incorrect trials. Each dot represents this correlation value for one recording session. The 11 sessions with both correct and incorrect trials for all 9 target locations were used. ($p = 0.004$, paired T-Test) *$p < 0.05$, **$p < 0.01$, ***$p < 0.001$.

incorrect: *median* = 71.4 ms. Wilcoxon Signed-Rank Test, $p = 0.001$) (Fig. 2g; Supplementary Fig. S3c, d). To obtain a more accurate measurement of the firing times standard deviation, we conducted an analysis including the 11 sessions with correct and incorrect trials for all nine remembered target locations. Again, the standard deviation of neurons' peak firing time ($n = 2051$) during correct trials (*mean* = 1358 ms) is significantly lower than incorrect trials (*mean* = 1828 ms; 1-way ANOVA, post-hoc, $p = 3.8E−09$). This suggests that increased temporal precision of firing peaks in single neurons influenced task performance.

## Time boundary cells parcellate trial periods

One may ask whether a single sequence encoded the entire trial in our task or whether the NASs were specific for the different trial periods. In the latter scenario, one would anticipate the existence of some neural signal that 'internally' parcels the trial period (e.g., cue from memory period, and memory period from navigation). Such time boundary cells have been reported during parcellation of long-term memory episodes in the human hippocampus[29]. We hypothesized that if there were NASs specifically triggered by the remembering of the target location, we should then find a signature signal that anticipates the beginning and end of the memory period. We found that a small subset of cells fired consistently at the same time in the sequence across trials of all 9 targets (*mean* = 4.8%, *median* = 4.7% of cells). The peak firing times of these cells therefore do not contribute to representing target, but rather encode a specific time within the trial. We observed that these time selective cells were active predominantly at two peak times preceding the onset of the memory delay and navigation epochs (Fig. 3a, e, see Methods - Time-Boundary Cells and Supplementary Fig. S5a–c for examples). Time boundary cells were active approximately 750 ms before the onset of the memory delay and 500 ms

before the onset of the navigation period (Fig. 3a, e), suggesting the animals anticipated the beginning of these events rather than reacting to externally triggered boundaries (e.g., cue offset and allowance of joystick movement). In the following analyses, we use the peak firing times of these time boundary cells to define neural boundaries dividing the task epochs.

## NASs are predictive of remembered targets and contain features of visual experience

Next, we examined whether NASs could encode the contents of working memory during the memory period of the task. We developed a computational method to analyze NASs in single trials and link them to behavior. We represented individual sequences of peak firing times during the memory period (Fig. 3a) defined by the time cell boundaries (Fig. 3e) in each trial across the population of recorded neurons as vectors of spike times (See Methods - Sequence Representation). Importantly, the NASs rely solely on temporal information about the timing of single neuron activations, independent of rate information that may be contained in the elevated activity. We performed dimensionality reduction on the matrix of correlations between NASs. The resulting component values are projected into a 3-dimensional space where each colored circle represents a cluster centroid for a different target condition (Fig. 3b). We also repeated the same correlation analysis using the memory period defined by the trial event timing (Supplementary Fig. S4a–c).

We found that condition centroids tend to group into three clusters that correspond to the three directions of the animals' movement trajectories through the virtual space (Fig. 3b). We hypothesized that the columnar grouping may relate to the view of trajectories within the virtual arena from the animal's perspective, in which the left-right (horizontal) dimension is expanded relative to the front-

rear (depth) dimension (Fig. 3c, d). This is a well-documented effect in which perceived distance in the depth dimension is less accurate than in the horizontal dimension from the viewer's perspective[30]. Indeed, the distribution of gaze positions in the lateral dimension was wider than in the vertical dimension[10]. Here, one must consider that although the virtual arena was designed in 3D with equal distances between targets in the horizontal and depth dimension (Fig. 3c), gaze is a 2-dimensional variable that compensates for the representation of 3D space and the viewer's angle. Thus, we used the approximate perspective projection of the maze obtained from the virtual layout in the video game engine (Fig. 3d) to obtain a realistic estimate of the target locations as seen by the animal from their viewpoint.

We then explored the direct relationship between NASs and behavior by calculating the Spearman correlation coefficient between matrices containing the Euclidean distance between condition centroids (Fig. 3f) and the Frechet distance between trajectories on the perspective view of the virtual arena (Fig. 3g). The Frechet distance between two trajectories is a measure of similarity that considers the location and ordering of the points along the trajectories[31]. Visually, the Frechet distance matrix reflects the grouping of trajectories following the columnar organization bias (left, center, and right) observed in the centroid distance matrix (Fig. 3f, g).

The centroid and Frechet distance matrices were significantly correlated compared to those obtained when shuffling the target locations across trials, suggesting that the discriminability between NASs parallels the discriminability between trajectories to targets held in working memory (observed: *median* = 0.50, shuffle: *median* = 0.32, Paired T-Test: $p = 6.31e{-}04$). Moreover, the relationship between NASs and target trajectories predicts whether information is successfully maintained during the working memory delay period, with higher correlations for correct than incorrect trials (Fig. 3h, neural boundaries, correct: *mean* = 0.52, incorrect: *mean* = 0.27. T-test, $p = 0.004$. See Supplementary Fig. S4c for task boundaries). We repeated the analysis using trajectories in a bird's eye (top) view of the maze, optimal trajectories (direct paths from start to target) and target locations (Euclidean distance between targets) and found that the correlation with the centroid matrix was in all scenarios significantly lower than in the previous analysis (Supplementary Fig. S4f). Taken together, these results demonstrate the unexpected finding that NASs during the memory period were specific to the subject's viewpoint, consistent with the idea that the subjects hold navigation content in their working memory that are specific to their visual experience.

To determine whether the NASs were specific to working memory, we repeated the analysis during a Perception-Navigation Control task (PNC). This task was identical to the working memory task, except the target remained on screen throughout the memory delay and navigation epochs, so the animals did not need to anticipate the beginning of the memory period nor represent the trajectories to the remembered location in working memory. We did not find neural boundaries parcelling the epochs in this task (Supplementary Fig. S6a). Further, the correlation between sequence centroids and trajectories was significantly higher during the memory delay epoch in the working memory task than in the perception control task (Supplementary Fig. S4d, Spearman Correlation; working memory: *mean* = 0.65 perception: *mean* = 0.38, T-Test, $p = 3.4e{-}04$). These results indicate that trial period parcellation by time boundary cells (that signal the beginning and end of the working memory period) was specific to the working memory task because the offset of the cue was specific to that task, and that NASs were more correlated to behavior during working memory than during the perception-navigation control task.

## NASs encode target row and column within specific subpopulations

We developed an unsupervised distance-based classifier, derived from our dimensionality reduction approach, to predict target condition from single-trial NASs (see Methods - Decoding Analysis). This classifier decodes target condition from single trial memory and navigation periods NASs with accuracy higher than chance level (Fig. 4d, All Cell Target, mean 20% above chance). Decoding accuracy is improved by leveraging subsequences of cells that are selective for row and column. By predicting row and column separately (Fig. 4d, Row, *mean* = 23% above chance, Column *mean* = 38% above chance), then combining this information to predict a target location (Fig. 4d Combined Target, *mean* = 29% above chance), decoding accuracy is improved significantly. (See Supplementary Fig. S7c for the same result using task boundaries.) The superior column decoding accuracy stems from the fact that the trials tended to be grouped by target column in the low-dimensional projections (Fig. 3b, Supplementary Figs. S4b and S7f).

## NASs are task period specific

One may argue that the observed NASs represent activation of neurons with mnemonic 'place fields' similar to sequential activity of place cells in the hippocampus[20–22]. Inconsistent with this idea, the NASs were differentiable between the different task periods (cue, memory and navigation), evidenced through classification analysis (*mean* decoding = 76%, *median* decoding = 87%, compared to chance (33%): T-Test, $p = 5.3e{-}08$) (Fig. 4a–c, Supplementary Fig. S7a, b).

One might suggest that NASs reflected motor planning during the memory period or neural replay of planned trajectories during the navigation period. If this were the case, one would anticipate NASs during the memory and navigation epochs from the same trial to be highly correlated, and that this correlation would be higher than that of memory and navigation NASs from different trials. This was not the case. In fact, 96% of memory period NASs were no more correlated to the same-trial navigation sequence than to navigation NASs from different trials (see Supplement - Additional Statistics). These results indicate that NASs in macaque LPFC represent remembered trajectories to target locations, and that such representation is specific to the working memory delay period of the task. The latter point makes the representation distinct from the sequences of activations during theta oscillations in the hippocampus, which represent local pieces of a trajectory and occur on the timescale of a single theta cycle[32]. LPFC contains different subpopulations of neurons that seem to be particularly active during working memory and navigation. These NASs, representing temporally structured, distinct sequences of activations in the population, may enable a 'mental subspace' that can represent episodic information independently of the sensory and motor signals occurring during the cue and navigation periods.

## NASs codes outperform persistent firing codes specifically in the virtual reality task

NASs may operate as a mechanism for working memory coding in our task specifically because the memoranda have varying complex spatiotemporal structure. To test this hypothesis, we conducted the same set of analyses exploring macaque LPFC single neuron temporal precision and population NASs in a classic oculomotor delayed response task (ODR) (Fig. 4e, f, Supplementary Fig. S8a). The task we used included 16 possible target locations. In this task, the animals fixate a dot on a blank screen, after which a peripheral target is flashed for a short time period. After the target offset, the animals continue fixating the dot for a few seconds while remembering the spatial location of the target. Upon the fixation dot offset, the animals make a saccade towards the remembered location to obtain a reward[33–35]. Saccades are ballistic movements that practically 'teleport' the fovea from the start to the end point without perception of traveling a path during eye movement[36]. Thus, the spatiotemporal complexity of the ODR task is substantially reduced relative to the VR navigation task.

Unlike the VR navigation task, when neurons were ordered by peak firing time during the ODR task, the patterns of activation were often disrupted or incomplete (Supplementary Fig. S8c), suggesting

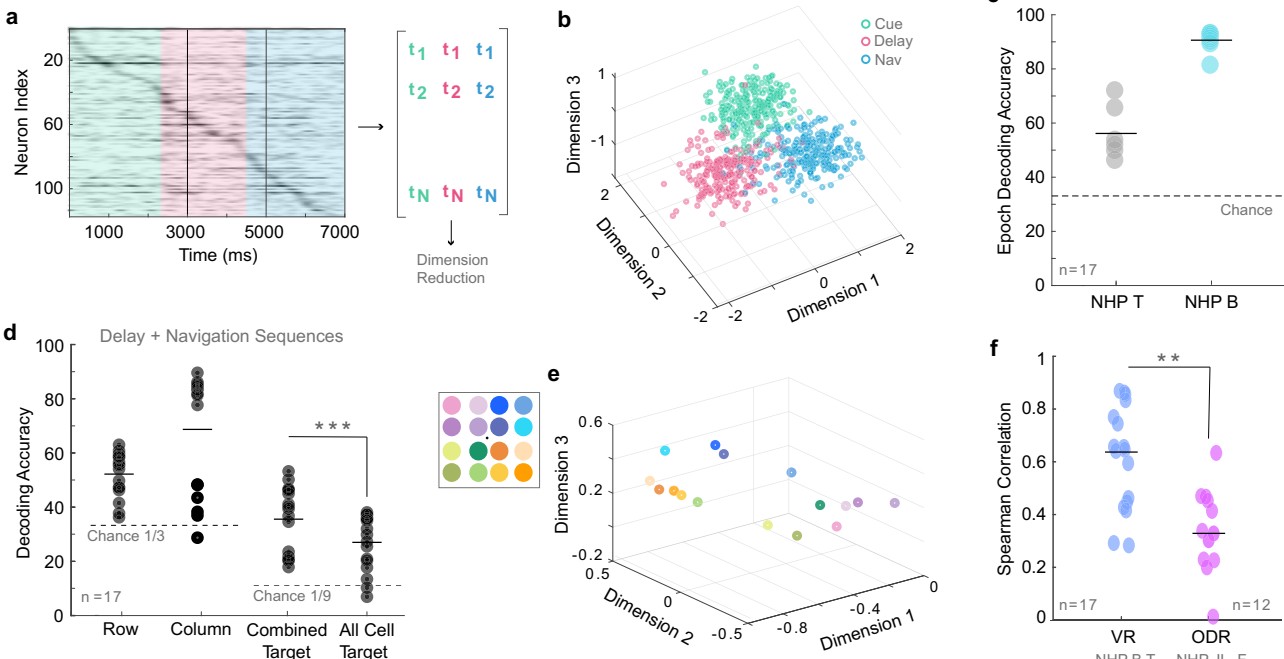

**Fig. 4 | NASs encode spatiotemporal structure of task during VR. a** NASs for the cue, delay, and navigation epochs were considered separately. Vertical black lines represent the task boundaries between epochs, and the colored panels represent the corresponding epochs defined by the neural boundaries. **b** A low dimensional summary of the data. Each point represents a sequence, colored by its epoch (as in '**a**'). **c** An unsupervised distance-based decoder classifies sequence epoch with high accuracy. **d** NASs from the delay and navigation epochs combined are used to decode memory content. Target row and column are decoded using subsequences of cells selective for row and column respectively. These row and column predictions uniquely specify 1 of the 9 target locations, and together produce a combined target prediction. This prediction improves decoding accuracy, compared to decoding the target location from the full sequence (all cell target). ($p = 7.8e{-}6$, paired T-test). **e** Dimensionality reduction analysis performed on data from an ODR task in NHP JL and NHP F with 16 target locations. **f** The relationship between neural sequence centroids and behavior in the task is significantly stronger in the VR task compared to the ODR task. Here, we consider the NASs during the delay period of each task. ($p = 0.002$, rank sum) *$p < 0.05$, **$p < 0.01$, ***$p < 0.001$.

that the organization of spiking activity in the ODR task may be different from the VR task. This may be due to neurons during the ODR memory period exhibiting less temporally consistent peak firing times from trial to trial. In many instances, real and shuffled distributions of firing time standard deviations were overlapping (Supplementary Fig. S8d). Indeed, the difference in means between real and shuffled distributions was significantly smaller in the ODR task compared to our naturalistic VR task (ODR1: *median* = 93.2. ODR2: *median* = 31.6, VR: *median* = 270.9; Kruskal–Wallis, $p = 1.2e{-}06$) (Supplementary Fig. S8e). Thus, NASs were inconsistent during the ODR task.

We next applied the dimensionality reduction analysis described in Fig. 3 to the ODR task data. Condition centroids were clustered in quadrants based on position of target location as reported previously using spike rate-based analysis[34] (Fig. 4e). However, when we calculated the correlation between the matrices of centroid distances and target locations, the correlation was significantly smaller in the ODR than in the naturalistic VR task (ODR: *median* = 0.34, VR: *median* = 0.64. Wilcoxon Rank Sum, $p = 0.0021$) (Fig. 4f). These results indicate that NASs are significantly more correlated to behavioral performance during the VR task than during the classic ODR tasks used in previous studies. The naturalistic VR task is different in several ways. First, it measures visuospatial working memory in a dynamic and more spatiotemporally complex environment. Second, it allows for free visual exploration via saccades. Third, it requires 3D navigation to a target location. These differences have clear impacts for the findings we report here (see Discussion).

A critical issue is whether the NAS code for visuospatial WM can outperform a target-selective persistent code. Specifically, the persistent code we consider here is comprised of the delay activity of cells that are (a) tuned for a particular stimulus throughout the delay period and (b) display elevated firing rates for their preferred targets compared to baseline. To address this question, we define subpopulations of cells in our dataset that display persistent delay activity, while progressively relaxing the criteria for persistence until the number of cells contributing to the persistent code nearly match the number of cells contributing to the delay period NASs (Fig. 5a, c). We then use a support vector machine (SVM) to make a preliminary comparison between the NAS and persistent firing codes. By estimating an SVM for the NASs and the set of persistent cells, we could directly compare accuracy for both potential codes using the same decoding algorithm. With this approach, we find that in the VR task, decoding accuracy for the NAS code is significantly higher than for the persistent code (Fig. 5b). During an ODR task performed by the same NHPs, however, we find the opposite: there are substantially more persistent cells compared to in the VR task (Fig. 5d, f), and the persistent code outperforms the sequence code in decoding ODR targets (Fig. 5e). Importantly, if we further restrict the criteria for persistent activity so that the fraction of persistent cells in ODR matches previous reports[35], we find no persistent cells in the VR task. This result suggests that, in the case of the naturalistic VR working memory task, the NAS code may provide a rich substrate for neural coding of working memory under spatiotemporally complex and changing sensory conditions.

Persistent firing codes have been consistently found to underlie encoding of WM items in ODR tasks[6,34]. In our ODR task, we find robust coding through persistent activity but we do not find NASs with a clear link to behavior – single neuron timing is unreliable, and the resulting unreliable sequences do not correlate with behavior. However, we find reliable NASs in our VR task that have a strong link to behavior, and that outperform the persistent code in decoding WM content. The

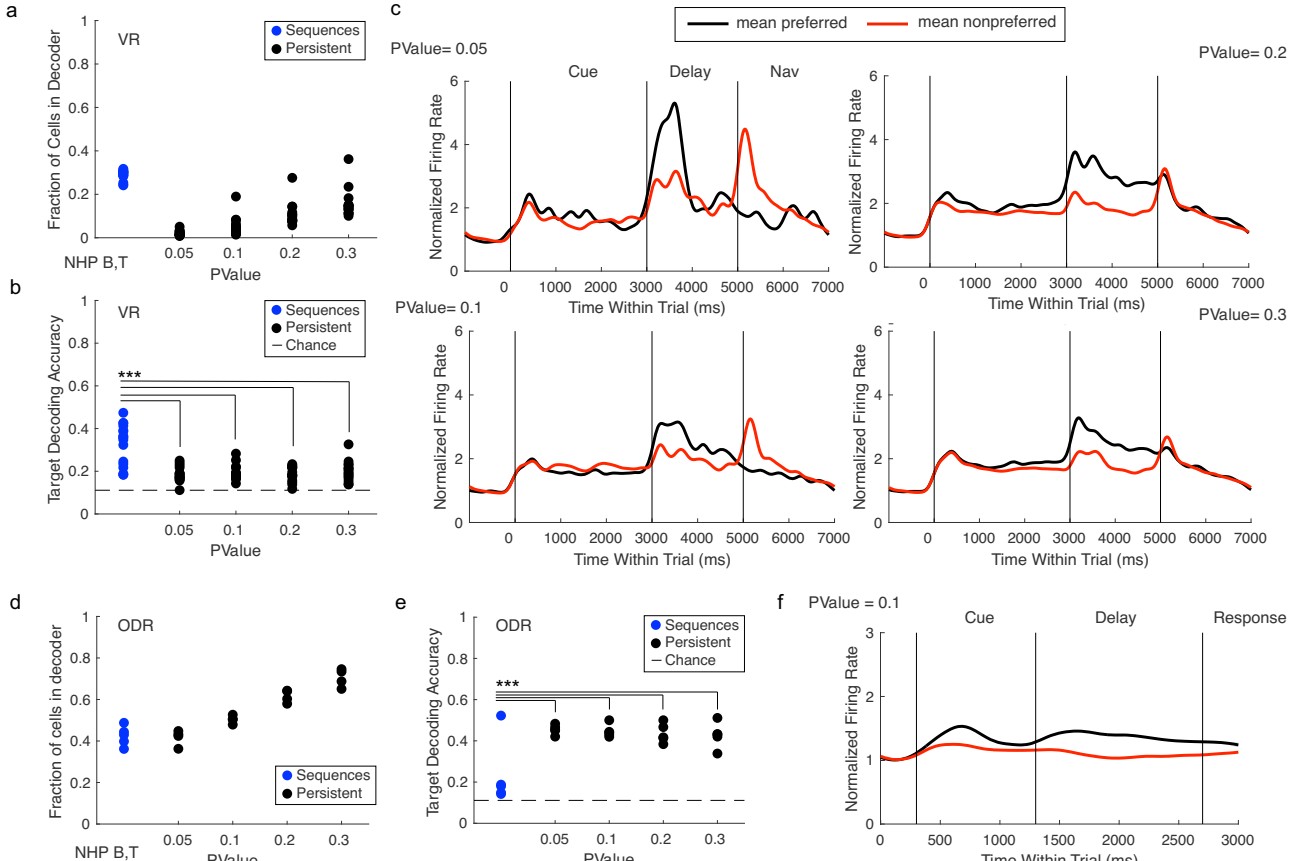

**Fig. 5 | NASs outperform the persistent code in the VR task. a** The fraction of cells that contribute to the delay sequence decoder (blue) compared to the persistent firing decoders (black). We progressively relaxed the ANOVA p-value criteria for defining persistent cells until the number of cells in the largest persistent decoder approaches the number of cells that contribute to delay period sequences. **b** 10-Fold cross-validated target-decoding performance of each decoder. (*p* = 3.8e−14, 1-way ANOVA, multiple comparisons corrected). **c** Average responses of persistent cells in each decoder for preferred targets (black) compared to non preferred (targets), with firing rates normalized to baseline. **d** Same as panel **a** but for the ODR task. (*p* = 0.001, 1-way ANOVA, multiple comparisons corrected). **e** Same as panel **b** but for the ODR task. **f** Same as panel **c**, *p* = 0.1 for the ODR task. *\*p* < 0.05, \*\**p* < 0.01, \*\*\**p* < 0.001.

ODR task constrains eye position during the memory period and requires saccade responses, which are stereotyped and ballistic movements. The VR task, on the other hand, requires a spatiotemporally complex response (virtual navigation), where eye position is unconstrained and therefore distractors and response strategies can be highly variable across trials (even of the same condition). Together, these results demonstrate that the LPFC circuitry can multiplex neural codes depending on the spatiotemporal features of the memoranda and task demands. This agrees with reports of representations in LPFC neurons that are flexible and diverse across different tasks[37–41]. Our present results indicate that such flexibility extends to working memory codes, where 'mental representations' may have diverse spatiotemporal features. These results thus have a substantial and fundamental impact on our understanding of WM in the brain.

**Ketamine disrupts NASs and impairs working memory performance**

In order to demonstrate a causal link between NASs and working memory in our naturalistic task, it was necessary to conduct a causal manipulation. We used ketamine, a N-methyl-D-aspartate (NMDA) receptor non-competitive antagonist that induces selective working memory deficits in humans and animals[26,42,43]. We injected subanesthetic doses of ketamine (0.25 mg/kg–0.8 mg/kg) intramuscularly while animals performed the task (see experimental timeline in Fig. 6a[26]). Ketamine drastically reduced performance of our virtual

working memory task b, colored vs gray dots). Working memory performance recovered 30 min to 1-h post-injection in the late post-injection period (Pre-Injection: *median* = 77%, Early Post-Injection: *median* = 28%, Late Post-Injection: *median* = 66%; Kruskal–Wallis, *p* = 8.5e−05) (Fig. 6b; Supplementary Fig. S9a, b).

After ketamine injection, there was a decreased difference in means between real and shuffled distributions of peak firing time standard deviations. This suggests that neurons fired with less temporal consistency after ketamine (Pre-Injection: *median* = 171.6, Early Post-Injection: *median* = 40.2, Late Post-Injection: *median* = 100.4; Kruskal–Wallis, *p* = 0.002) (Fig. 5c, Supplementary Fig. S9c, d). Behaviorally relevant groupings of condition centroids were similar between the non-injection data set and the pre-injection ketamine data set (Fig. 3b, Fig. 6d). This grouping was lost after ketamine injection but was regained 1h later as behavioral performance recovered (Fig. 6d). We also saw that the correlation between condition centroid distances and target trajectory distances decreased after ketamine and then recovered, indicating that NASs were less predictive of remembered target location immediately after ketamine injection (Pre-Injection: *mean* = 0.48, Early Post-Injection: *mean* = 0.26, Late Post-injection: *mean* = 0.39. 1-way Anova, post-hoc, *p* = 0.004) (Fig. 5e, Supplementary Fig. S9e, f, g). There was no change in any of the described measures in a saline control condition (Supplementary Fig. S9h, i). These results indicate a causal link between NMDA receptor dysfunction caused by ketamine and disruption of NASs leading to deficits in working memory.

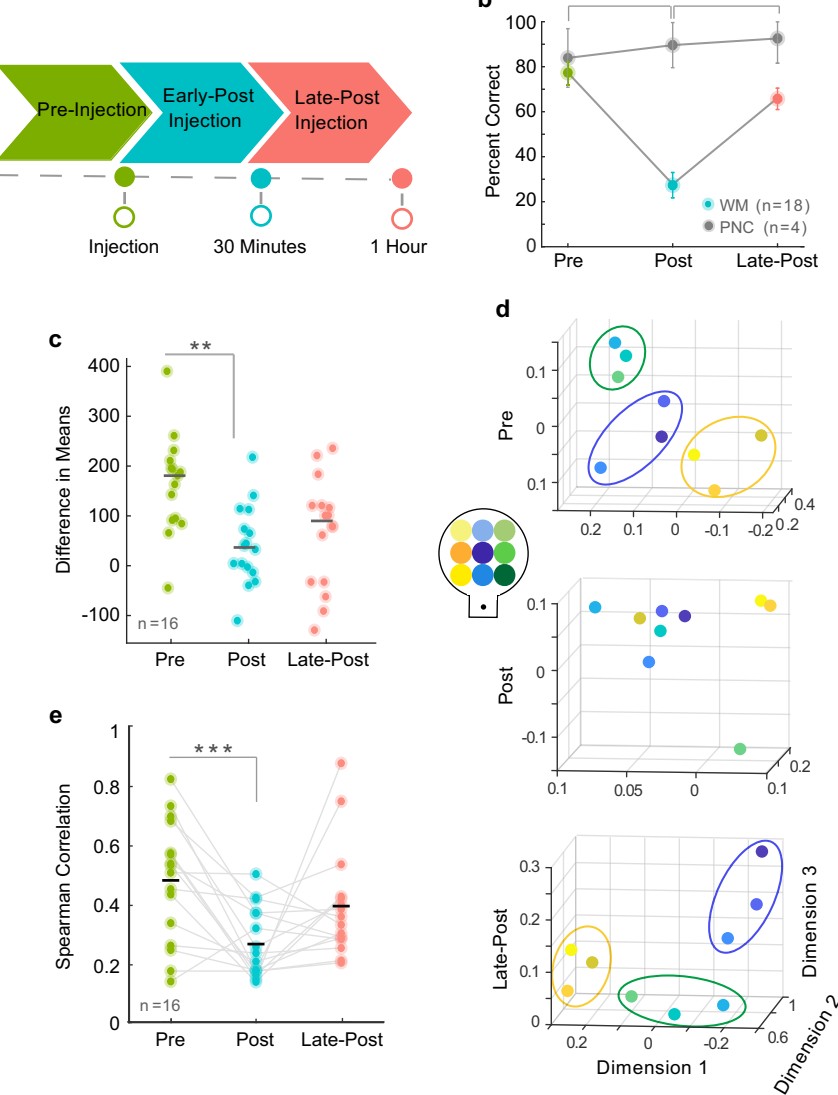

**Fig. 6 | Ketamine manipulation disrupts neural sequences and working memory. a** Experimental timeline for ketamine injection. Pre-injection period is depicted in green, early-post injection in blue, and late-post injection (recovery) in pink. **b** Task performance as percent of correct trials for each injection period. Color dots represent median values per injection period for working memory data and gray dots represent median values per injection period for perception-navigation control data. Asterisks indicate significance between working memory injection periods. Data is presented mean +/− SD. **c** Difference in means between distributions of standard deviations for real and shuffled spike-times, plotted for each ketamine injection period. Colored dots correspond to recording sessions and gray lines indicate means. **d** Condition centroids projected in 3D space. Centroid colors correspond with their position in the virtual environment (see inlet). Ellipsoids are illustrated guides to indicate behaviorally relevant groupings of targets in the pre-injection and late post-injection periods. These groupings are notably absent in the post-injection period. **e** Spearman correlation values for each ketamine injection period. Dots represent individual sessions and matching sessions are connected by lines. *$p < 0.05$, **$p < 0.01$, ***$p < 0.001$.

## Discussion

We recorded the responses of hundreds of single neurons in the macaque LPFC during a complex visuospatial working memory task set in a naturalistic virtual environment, searching for the neural correlates of the episodic memory buffer[44]. We report four major findings: (1) time boundary cells within the LPFC parcellate trial events during a cue-working memory-navigation task; (2) NASs during the working memory period encode specific targets in this WM task, in an integrative manner that goes beyond simple target location and includes subjective visual experience; (3) the NAS code outperforms the persistent code in the VR task, but not during a classic ODR task where a robust persistent code performs best and, (4) NMDA receptor antagonism induced by ketamine disrupts NASs, selectively impairing working memory performance.

## An internal code for representing the spatiotemporal structure of working memory

Our study reveals a mechanism for encoding working memory representations with variable spatiotemporal structure. We found time cells that anticipate the different events during the trial, parcellating the trial structure and providing temporal anchor points for the NASs. We propose time cell activation may be extremely important for triggering working memory NASs. Time boundary cells signal the beginning and end of the working memory period, effectively parcellating the mental operation that the animals need to perform (via NASs) to remember an on-screen path towards a location. Similar boundary cells have been found to exist for the parcellation of long-term memory episodes in the human hippocampus[29]. Thus, time boundary cells may be a general mechanism for parcellating spatiotemporal episodes defined by NASs.

The persistent firing hypothesis has been prevalent in the working memory field since the initial study by Fuster and Alexander[4,5]. Within this view, working memory content is encoded by neurons that selectively and persistently fire when remembering a certain item or location. A shortcoming of the persistent firing hypothesis is that a persistent rate code may not be most efficient to support working memory representations with diverse spatiotemporal structure[9,45,46]. Indeed, in tasks during which sequences of multiple items need to be held in working memory, persistent firing is scarce[9]. A recent study reported that during a multi-item spatial working memory task in which monkeys had to remember a series of spatial locations and make saccades to them in sequential order, temporally organized neuronal populations represented the order in which saccades were made[47]. Although these studies did not specifically test animals in a complex task comparable to that of our VR task (i.e., unrestrained eye movements, rich dynamic visual scenery and navigation using a joystick), they suggest that persistent firing is not a monolithic mechanism generalizable across working memory tasks.

Our paradigm differs from those used in previous studies. We did not use sequences of different memoranda; instead, our task trials were fluid and required subjects to hold complex, spatiotemporal information in working memory, consistent with Baddeley's idea of temporal episodes. Subjects remembered a single target location and the trajectory to the location in a 3D virtual naturalistic environment. Importantly, our study did not restrain eye position, allowing for naturalistic exploration of the scene while information was being held in working memory. The rationale behind studies restraining eye position is to avoid the interference caused by eye position signals, changes in the retinal image and, consequently, in visual inputs on the working memory representation[48]. However, in naturalistic conditions, working memory coding must be robust to such changes. Indeed, we have recently demonstrated that changes in eye position during exploration of virtual environments cannot account for the observed changes in population activity during encoding of different task elements in the LPFC[10].

Previous studies in macaques have proposed that transiently active neurons maintain representations through shared temporal relationships. However, they were unable to record large numbers of simultaneously active neurons and thus to isolate NASs[49]. Our study has overcome this limitation by recording from hundreds of simultaneously active neurons, revealing precise sequences of single unit spiking activity that encode specific working memory content. Studies in mice that simultaneously record from many neurons have reported NASs during short-term memory tasks in the posterior parietal cortex and dorsomedial striatum[19,50]. In the rodent hippocampus, sequences of place cell activation signal trajectories to remembered locations that are stored in long-term memory[32]. Thus, sequence codes seem to be utilized for re-playing episodic memories across species.

However, the NASs we report in this study differ in several ways from those described in previous studies. First, they occur in the LPFC, a brain area that appears de novo in anthropoid primates[3]. More specifically, the NASs reported here occur within the supragranular layers 2 and 3, where working memory representations are found[51,52]. The expansion of neocortical layers 2/3 in anthropoid primates is accompanied by changes in the morphology, size and intrinsic properties of pyramidal cells[53], as well as changes in the proportion of interneuron types that regulate the balance between inhibition/excitation of pyramidal cells[54] relative to other species and brain areas. The LPFC may have evolved a microcircuitry for holding various types of working memory codes and multiplex them according to the spatiotemporal structure of the memoranda, e.g., via persistent firing when the memoranda is invariable over time[55], and via NASs when the task requires responses with more complex spatiotemporal structure, as we show here. Such versatile mental representations can be dissociated from sensory and motor signals and may be key to an enriched mental world that enables the enhanced cognitive control, planning, and creativity observed in anthropoid primates [3].

Through pharmaceutical manipulation, we identify that the generation of working memory related NASs seems to depend on NMDA receptor function, though other neurotransmitter systems may also be involved. The interaction between inhibitory interneurons and excitatory pyramidal cells plays an important role in LPFC prefrontal circuits during working memory tasks[56]. Therefore, the precise activation of pyramidal cells may be dependent on a temporally coordinated 'release of inhibition' by interneurons[57,58]. A recent work from our group studying the same task and NHPs demonstrated that ketamine at low doses selectively decreases the firing of narrow spiking interneurons, likely mediated by GluN2B-containing NMDA receptor dysfunction[26]. The working memory deficits induced by ketamine were accompanied both by decreased firing of narrow spiking inhibitory interneurons and increased firing of excitatory cells. These findings align with a previously proposed mechanism for working memory disfunction - that reduced NMDAR conductance of interneurons leads to generalized disinhibition of pyramidal cells[59–61]. In classic WM tasks, this results in a loss of tuning, and therefore of spatial specificity of WM representations. Here, a parsimonious explanation for our findings is that ketamine induced a loss of firing in narrow spiking interneurons (e.g., PV basket of chandelier cells), which in turn impaired their ability to coordinate NASs in pyramidal cells, ultimately disrupting the sequences and causing deficits in episodic working memory. The fact that the effect of ketamine was selective for the working memory task supports the view that the sequential activation mechanism reported here is particularly important for supporting mental representations that 'live' within the LPFC microcircuits.

One may ask, what are the benefits and potential drawbacks of a sequence-based code relative to other coding schemes? Some aspects of sequential codes have been explored in computational and theoretical work, such the mechanisms by which downstream neurons can learn to detect and decode sequential activity[62]. However, there are still open theoretical questions, such as how a sequence code could handle a variable length delay period: could additional cells be recruited or might the timing of the sequences adapt depending on features of the task? For a single neuron, sequence codes would be more energy efficient than codes based on persistent firing since neurons increase firing only during a certain time interval and then return to their baseline rates. We speculate that a sequence code may also provide working memory systems with the temporal resolution to encode varying spatiotemporal information, such as changes in 3D scenery involved in navigating to a target in the virtual reality task. Remarkably, the correlation between NASs and target trajectories during correct trials remains stable even after removing 70% of neurons from the population. 80 − 90% of neurons must be removed for this correlation to significantly change, at which point the correct trial correlation equals the incorrect trial correlation (Supplementary Fig. S4e). Thus, the sequence code seems to be robust to losses in the number of neurons. The NAS code is also robust to trial-to-trial variations in the firing rate in individual neurons[63] since it relies on the timing of the activation rather than on spike counts of individual neurons. The fact that sequence codes are found in multiple species and brain areas to represent distinct types of varying spatiotemporal information may point to an evolutionary preserved mechanism intrinsic to brain circuits. The expansion of the LPFC in primates may allow the coexistence of such a mechanism and persistent firing to support flexible and complex mental representations.

Our results demonstrate a code in the primate lateral prefrontal cortex, a potential candidate for encoding the rich spatiotemporal structure of working memory episodes. NASs can bind working memory information in space and time, independently of sensory cues and motor signals, and without necessarily undergoing long-term memory storage. NASs are sensitive to ketamine, suggesting that

NMDA receptor activity plays a role in their neural dynamics. The parallel of NASs occurring on the timescale of single theta-cycles in the hippocampus[32] with the seconds-long, behavioral timescale NASs we report here, represents an intriguing possibility to connect short-and long-term memory systems throughout the dynamics of the LPFC.

## Methods

We used the same two adult male rhesus macaques (*Macaca mulatta*) in the main experiment as well as the ketamine and saline experiments (age: 10, 9; weight: 12, 10 kg). The oculomotor delayed response task labeled ODR1 was recorded from the same animals, while ODR2 was recorded from two different male macaques using one multielectrode Utah array implanted in each animal [33,34].

### Ethics statement

Animal care and handling (i.e., basic care, animal training, surgical procedures, and experimental injections) were pre-approved by the University of Western Ontario Animal Care Committee. This approval ensures that federal (Canadian Council on Animal Care), provincial (Ontario Animals in Research Act), regulatory bodies (e.g., CIHR/NSERC), and other national standards (CALAM) for the ethical use of animals are followed. The oculomotor delayed response task experiment complied with Canadian policies and regulations and was pre-approved by the McGill University Animal Care Committee[33,34]. Regular assessments for physical and psychological well-being of the animals were conducted by researchers, registered veterinary technicians, and veterinarians.

### Task description

Each trial begins when the virtual environment appears on screen with one target location cued (Fig. 1d, red vertical bar). The cue remains on screen for seconds. The disappearance of the cue is followed by a 2 s memory delay period. During both the cue and memory periods (or, epochs), the subject is free to visually explore the space, but navigation is disabled. After 2 s, navigation is enabled, and the subject navigates to the remembered location using a joystick. The subject correctly completes a trial by reaching the target location. (See Supplementary Movie 1) They are not required to stop at the target, but only a small radius of space will trigger a correct response and a juice reward (1/2 grape-apple juice, 1/2 water). The responses take varying amounts of time depending on the trial and the target condition; however, only 1-2 s of navigation were included in the analysis. This length of time was chosen trial-by-trial to ensure the subject had not yet reached the target location and received a reward. During the trial, if after 10 s of navigation the subject has yet to reach the target location, the trial times out and is considered incorrect. At the completion of a trial, the subject is virtually teleported into a black box (blank, black screen) to await the next trial. The inter-trial-intervals are variable, but around 500 ms on average. No additional visual or auditory cues accompany the reward or boundaries between task epochs.

The virtual task environment was developed using Unreal Engine 3 development kit, utilizing Kismet sequencing and UnrealScript (UDK, May 2012 release; Epic Games). Details about this platform and the recording setup can be found in ref. 64. Movement speed through the environment was fixed. Target locations within the virtual arena were arranged in a 3 × 3 grid and spaced 290 unreal units apart (time between adjacent targets is approximately 0.5 s). The perception-navigation control variation of the task (PNC) was identical to the working memory version except that the targets remained onscreen through the trial. In this control, navigation was guided by perception, and WM was not required.

### Experimental setup

Animals performed the task in an isolated room with no illumination other than the monitor. The room contained no AC power lines and

was radiofrequency (RF) shielded. The task was presented on a computer LDC monitor positioned 80 cm from the subjects' eyes (27" ASUS, VG278H monitor, 1024 × 768 pixel resolution, 75 Hz refresh rate, screen height equals 33.5 cm, screen width equals 45 cm). Eye positions were monitored using a video-oculography system with sampling at 500 Hz (EyeLink 1000, SR Research). Stimulus presentation was controlled through a custom computer program (through Unreal Engine 3). Subjects were seated in a standard enclosed primate chair (Neuronitek) during the experiment and were delivered juice through an electronic reward integration system (Crist Instruments). Prior to the experiments, subjects were implanted with custom fit, PEEK cranial implants which housed the head posts and recording equipment (Neuronitek). See ref. 65 for more information. The head posts were attached to the primate chair for head fixation.

The experimental setup for the oculomotor delayed response task is outlined in both[34,35].

### Ketamine injection

The ketamine doses were titrated so they did not induce visible behavioral changes in the animals (i.e., nystagmus or somnolence). An intramuscular injection of ketamine (Narketan, 0.25, 0.4, or 0.8 mg/kg) was administered in the hamstring muscles by a registered veterinary technician. Ketamine injections were spaced at least two days apart to allow for washout of the drug. Saline administration was conducted identically with a fixed 0.25 mg/kg dose (See ref. 26 for more details).

### Surgical procedure

Custom PEEK implants which housed recording hardware and a headpost were developed and implanted in each animal[65]. Brain navigation for surgical planning was conducted using Brainsight (Rogue Research Inc.) (Supplementary Fig. 10a). Two 10 × 10, microelectrode Utah arrays (96 channels, 1.5 mm in length and separated by at least 0.4 mm) (Blackrock Neurotech) were chronically implanted in each animal. Electrodes were implanted in the left LPFC (anterior to the arcuate sulcus and on either side of the posterior end of the principal sulcus) (Supplementary Fig. 10b, c). Arrays were impacted approximately 1.5 mm into the cortex. Reference wires were placed beneath the dura and a grounding wire was attached between screws in contact with the pedestal and the border of the craniotomy. Surgical procedures were conducted under general anesthesia induced by ketamine and maintained using isoflurane and propofol.

For the oculomotor delayed response task data, a 96-channel Utah array was implanted in each monkey's left LPFC in the same region that electrodes were implanted for recording during performance of the virtual working memory task (Supplementary Fig. 8b). Detailed surgical methods can be found in [34,35].

### ODR task details

**ODR 1:** The oculomotor delayed response task performed by NHP B and NHP T was separated into four epochs: fixation, stimulus presentation, delay, and response. The fixation period duration was 300 ms (NHP B) or 500 ms (NHP T). The cue was presented for 1000 ms. The cue then disappeared and the subject maintained fixation for an additional 3000 ms (NHP B) or 1500 ms (NHP T), after which the fixation point disappeared and the subjects responded by making a saccade to the remembered location.

**ODR 2:** The oculomotor delayed response task performed by NHP JL and NHP F was also separated into four epochs: fixation, stimulus presentation, delay, and response. The animal began a trial by fixating on a fixation dot and by pressing a lever. The duration of the fixation period was either 482, 636, or 789 milliseconds. A sine-wave grating target then appeared at 1 of 16 randomly selected locations positioned in a 4 × 4 grid for 505 ms. This was followed by a delay period ranging from 494–1500 ms. The fixation point was removed, cueing the animal

to make a saccade to the location of the previously presented target and then to release the lever (see[34,35] for more details).

## Analysis

**Task performance.** Correct trials are trials in which the animal reaches the correct target location within 10 s. An incorrect trial occurs if the animal does not reach the target location within 10 s. Percent of correct trials is calculated as the number of correct trials divided by the total number of trials. Response time (Supplementary Fig. S1a) was calculated for correct trials as the time from the start of navigation to the time in which the animal reaches the correct target location.

The optimal trajectory analysis (Supplementary Fig. S1b) was calculated for correct trials. It is calculated as the real length of the animal's trajectory to correct target location divided by the length of the optimal trajectory (i.e., the Euclidean distance from the start position to the target location).

For incorrect trials, we calculated the distance from the animal's final position to the correct target location (Supplementary Fig. S2d). Distance values were modified from arbitrary 'Unreal' units (the unit system in Unreal Engine Development Kit, Unreal Engine 3, Epic Games) to 'Unreal' units divided by the distance between two targets to increase interpretability. A new value of 1 would represent 290 unreal units (the distance between two adjacent targets).

**Eye behavior.** Percent of eyes on screen (Supplementary Fig. S2a) measures the number of eye data points falling on the screen divided by the total number of eye data points. Off screen data points occur when the animal looks off screen or closes their eyes (as occurs during blinking).

Eye data was classified into fixations and saccades based on a method outlined in[66] that was developed for use in a similar virtual environment. The percent of fixations on target (Supplementary Fig. S2c) was calculated by the number of fixation events falling within a trial's target location divided by total number of fixation events. We used a linear classifier (SVM) (Libsvm 3.14[67]) with 5-fold cross validation to predict target location from eye fixation position data (Supplementary Fig. S2d, e).

The main sequence (Supplementary Fig. S2f) was calculated by separating saccades into bins of 3° of amplitude, starting at 2° and computing the medians for each bin. The proportion of single units tuned for eye position in both retinocentric and spatiocentric reference frames was calculated using a quadrant binning pattern for a $40° \times 30°$ field (Supplementary Fig. S2g). A bin had to have at least ten saccades to be acceptable and sessions had at least three acceptable bins.

**Spike processing.** Neuronal data was recorded using a Cerebus neuronal Signal Processor (Blackrock Microsystems) via a Cereport adapter. The neuronal signal was digitized (16 bit) at a sample rate of 30 kHz. Spike waveforms were detected online by thresholding at 3.4 standard deviations of the signal. The extracted spikes were semi-automatically resorted with techniques utilizing Plexon Offline Sorter (Plexon Inc.). Sorting results were then manually supervised. Multi-units consisted of threshold-crossing events from multiple neurons with action potential-like morphology that were not isolated well enough to be classified as a well-defined single unit (for spike sorting example see Supplementary Fig. 10d, e). We collected behavioral data across 20 working memory sessions (eight in animal T, twelve in animal B) and neural data across 17 sessions. This yielded a total of 3950 units recorded: 2578 single neurons (346 in animal T, 2232 in animal B) and 1372 multiunits (512 in animal T, 860 in animal B). We collected behavioral data across 18 ketamine-working memory sessions (nine in animal T, nine in animal B) and neuronal data from 17 ketamine-working memory sessions with one session from animal T removed due to incomplete synchronization of neuronal data during the

recording. This yielded a total of 2906 units recorded during ketamine-working memory sessions: 1814 single neurons (259 in animal T, 1555 in animal B) and 1092 multiunits (533 in animal T, 559 in animal B).

**Spike density function.** Spike density functions (SDFs) were generated by convolving the spike train with a Gaussian kernel (standard deviation = 100 ms).

**Time consistent neurons.** To quantify time consistency of neurons, we created SDFs combined between electrode arrays over the entire trial time using neurons with firing rates above 0.5 Hz. SDFs were created for each condition that contained at least five trials. We calculated the peak firing time for each neuron in the population, then calculated the standard deviation of the peak firing time for each neuron over all trials in a condition. Finally, we created a probability distribution from the standard deviation values (Fig. 2e). Correct and incorrect trials were considered separately. We then shuffled the peak firing times for each neuron from trial to trial and created a shuffled probability distribution. We calculated the difference in mean values between the real and shuffled distributions to get the mean difference value (Fig. 2f, g). To calculate the standard deviation values plotted in Supplementary Fig. 3b, we calculated trial-trial standard deviation of peak spike time for the target condition in which each neuron fired the most consistently during correct trials (i.e., lowest deviation). The same conditions were used for shuffled data and for incorrect trials.

We repeated the same analysis for both ODR tasks (Supplementary Fig. S8d,e) with the following slight variations:

ODR 1: This data was collected from the same animals and electrodes as our naturalistic VR task. This task contained 16 targets (Supplementary Fig. 8a) and was a variation of a traditional ODR task in which the fixation point changes location across trials, resulting in many different task conditions with varying combinations of fixation location and target location. For this reason, we grouped trials with target locations within the same quadrant (same direction saccade). To match the task structure of the VR task, we did not use data from the fixation period.

ODR 2: This data was collected from NHPs JL and F (Supplementary Fig. 7b) using one Utah array implanted in the left LPFC (same region as NHP B and T). This task contained 16 targets with a consistent central fixation point (Supplementary Fig. 8a). To match the task structure of the VR task, we did not use data from the fixation period. Since the ODR2 task had jittered delay epoch timing, we used trials with delay periods >1000 ms and included the first 1000 ms of the epoch.

**Sequence representation.** Each trial was represented as a vector of spike times, **S**, with each component given by the time of maximum spike density of the corresponding neuron. For a recording session with N neurons and T trials, this process results in a set of T vectors of length N. Let $t_n$ represent the within-trial time t at which neuron n reached its maximum spike density. Then $\mathbf{S}_n = t_n$. When the neurons were sorted in increasing order of $t_n$ values, clear bands of elevated spiking activity are visible which span both arrays and recording length (Fig. 1g, h). While it may be possible to define sequences in shuffled or random data by virtue of sorting, we note that the sequences we observe cannot be obtained by chance. The bands of elevated firing are not observed in shuffled data (Supplementary Fig. S3a). Further, the sequences consistently span both the full trial and array, tiling the rasters with only one of many possible slopes.

**Time-boundary cells.** Cells were considered time-selective if they contributed to the sequence at the same time, $t_n$, across trials of all 9 target conditions (i.e., their position in the sequence is consistent across trials and not related to target condition - see Supplementary Fig. S5a, b for examples). To quantify this for each neuron, we

computed the full-width half-max of the distribution of spike times across all trials in a recording session (see Supplementary Fig. S5c). If the width fell below a threshold, that cell was labeled time-selective. We used thresholds of 1 s for NHP B and 2 s for NHP T to obtain similar fractions of time-selective cells across subjects; however, we note that a 2D scan across thresholds (height and width) suggests the specific thresholds chosen does not significantly impact results. For example, using a threshold of 1.5 s for both subjects does not change the location of the peaks.

These time-selective cells were then used to define neural boundaries. We pooled the spike times $t_n$ of all time-selective cells across all recording sessions for each subject. The histograms of these spike times reveal two clear peaks, anticipating the delay and navigation epochs (Fig. 3e). These peaks represent the times within the trial at which time-selective cells contribute most often to single-trial sequences. The peak times were used to define the neural time boundaries used in the following analyses. Note that no neural boundaries were found for the perception-navigation control task (Supplementary Fig. 6a), so the task boundaries were used for comparisons to that control task (Supplementary Fig. 4d).

**Epoch-specific sequences.** To consider a sequence specific to a given task epoch, the contribution of any cells with spike time $t_n$ outside the desired epoch were set to NaN. In the main analysis, the neural boundaries were used to define these epochs (Figs. 3–5). However, results are not significantly impacted if the task boundaries were used to define the epochs instead (Supplementary Figs. S4a–c, S7a–c and S9g).

**Dimensionality reduction.** For each recording session with T trials, a correlation matrix was created by computing the Pearson correlation coefficient between each pair of vectors $S_i$ and $S_j$ representing sequences for trials i and j. Let X be the resulting $T \times T$ correlation matrix. This matrix captures the similarity of neural sequences across trials. High-valued blocks of this matrix represent groups of of trials in which similar sequences occurred.

The correlation matrix X was then projected onto the eigenvectors corresponding to the eigenvalues with largest modulus to generate a low-dimensional summary of the data in 3D-space. This approach is similar to spectral clustering on the correlation matrix; however, rather than performing kmeans clustering on the eigenvectors, we treat them as axes of a similarity space into which we project the matrix of correlation values. The details of this projection are as follows:

The eigenvalues $\lambda_1 ... \lambda_T$ of X with corresponding eigenvectors $v_1 ... v_T$ were computed, and labeled such that $|\lambda_1| \geq |\lambda_2| \geq ... \geq |\lambda_T|$. Note that since X is a real-valued symmetric matrix, all eigenvalues are real.

Let Q be the matrix with columns given by the first three eigenvectors of X, $Q := [v_1, v_2, v_3]$. Define the $T \times 3$ projection matrix $P = XQ$. This projection matrix is used to define a similarity space in which the point $(P_{(j,1)}, P_{(j,2)}, P_{(j,3)})$ describes the (x,y,z) coordinates of a representation of one trial j in 3-space.

The points in this projection each correspond to one trial, and their positions are determined by the relative similarity of the corresponding sequences. The centroids of the clusters corresponding to each trial condition were then determined, and the matrix of Euclidean distances between the centroids was computed. (See Supplementary Fig. 7e for an example.) In this way, the spiking data from each recording session was reduced to a $9 \times 9$ distance matrix (Fig. 3f).

We note that considering higher dimensional projections does not significantly increase cluster separation in the projection (see Supplementary Fig. 7d).

**Correlation analysis.** Visuospatial trajectories were computed for each trial by transforming the movement trajectories (in a 3D Unreal coordinate system) into 2D screen coordinates using a perspective projection matrix. Average visuospatial trajectories were computed for each recording session by averaging the screen coordinates of correct trial paths to each target in the virtual environment (excluding outlier trajectories with z-score > 1 of mean Frechet distance to other trajectories in the group).

A $9 \times 9$ distance matrix was then created for each recording session by computing the Frechet distance between each pair of average visuospatial trajectories[31] (See Fig. 3d for example trajectories and Fig. 3g for example distance matrix). Intuitively, the Frechet distance can be understood as the shortest leash required to walk your dog if you followed one trajectory and your dog followed the other. This distance measure reflects differences in the curvature and separation between trajectories as well as the end positions. Further, since trajectories are computed from the behavior during each session separately, it reflects behavioral differences between subjects and differences across days.

The Spearman correlation coefficient between the trajectory distance matrix and the centroids distance matrix was computed for each recording session. This provides a measure for how much the organization of the centroids in 3-space reflects the behavior during the task. To compare correct versus incorrect trials (Fig. 3h, Supplementary Fig. 4c), and WM versus the perception-guided navigation control (Supplementary Fig. 4d), the relevant groups of trials were used to define separate sets of centroids. The correlation analysis was performed for each set of centroids, and the resulting correlation values were compared.

This procedure was repeated for several different aspects of the task structure, to determine which was best reflected by neural sequence structure (see Supplementary Fig. 4f). Average movement or World trajectories are trajectories in the 3D Unreal coordinate system, before the perspective projection was applied to create the visuospatial, or Screen trajectories. Optimal trajectories refer to the shortest path from the start location to each target location. Target location refers to the position of the target in the virtual arena. In all cases, the distances between the 9 conditions were computed using the relevant distance measure (Frechet for trajectories, Euclidean for target locations), to create the $9 \times 9$ distance matrix used in the analysis.

**Ablation test.** A percentage of cells (from 10 to 90 percent) were randomly excluded (i.e. the corresponding sequence components $S_j$ were set to NaN), and the correlation analysis was performed. This was repeated for 100 iterations at each percent of cells removed. The average correlation across iterations was reported (Supplementary Fig. 4e).

A similar analysis was used to compare the sequence contributions of cells that are classically tuned for target location during the delay versus untuned cells. Here, tuned and untuned population sequences were considered separately by setting the opposite population to NaN, and performing the correlation analysis. Tuned and untuned populations of matched size were considered: the population with fewer cells defined the size, N, of the matched population, and N cells were randomly selected from the other population to define the matched sequences. This random selection was repeated for 100 iterations and the resulting correlations were averaged (see Additional Statistics).

**Decoding analysis.** All decoding analyses from Fig. 4 (trial epoch, target location, target column, target row) were performed using variants of a simple classifier based on distances within the low-dimensional projection of the data described above (see Dimensionality Reduction for projection details). Decoding proceeded as follows:

The data set to be decoded is the set of points in 3-space, $\{P_j = (x_j \ y_j, z_j)\}$, corresponding to the set of trials j in a given recording session. These data points have ground-truth labels $\{g_j\}$ (which could

be trial epoch, target location, target column, or target row depending on the application).

Training and test sets were determined using 5-fold cross-validation. For each of the 5 iterations, the training set was used to determine condition centroids (i.e.,the coordinates of points $P_j$ corresponding to the same target were averaged, following the procedure outlined in Dimensionality Reduction). The distances between each point $P_j$ in the test set and each condition centroid were then computed. Each trial j was assigned a label $c_j$ according to the condition that minimized this distance. Decoding accuracy was computed as the fraction of points in the test set assigned to the correct condition (i.e., the fraction of trials for which $c_j = g_j$), and the average accuracy across the 5 iterations was reported.

Since the 9 targets in the task are positioned as a $3 \times 3$ grid in the virtual arena, each (row, column) pair defines a target location. In Fig. 4d, combined target decoding was performed by first decoding the target row and column (as described above), then using the predicted row and column to determine which of the 9 target locations each trial corresponds to. (Note that in this case, only row-selective and column-selective cells were included in the sequences to decode row and column respectively. See below for definition of selectivity.) In the same figure, All Cell Target decoding was performed as described above, by predicting the target location directly from the full sequences.

**Persistent firing analysis.** An additional decoding analysis was performed to provide a preliminary comparison between persistent coding and the NAS code in this data. Cells are considered to display persistent delay activity if they satisfy two criteria: (1) the cell must display target-selective tuning during the delay period and (2) the cell must display a significantly higher firing rate for tuned targets during delay relative to baseline (with baseline rates determined from the inter-trial intervals). Both criteria are determined using a *p*-value threshold from and ANOVA, and this threshold is progressively relaxed to include more cells in the persistent population. With this definition, we find a set of persistent cells in both the VR and ODR task (sessions for NHP B and NHP T), consistent with previous literature. We note that the persistent population in the ODR task is somewhat larger than previously reported, and that a stricter p-value threshold can reduce this population to a more standard size, but when doing so, we no longer obtain any persistent cells in the VR task.

We then train two support vector machine (SVM) decoders. The first predicts trial condition using the delay activity of cells identified as persistent. The second predicts trial condition using the coordinates of the sequence projection (see Dimensionality Reduction for details). Both are cross-validated (10-fold).

**Determining neural selectivity.** Neural selectivity was determined from the set of times at which each cell contributed to the population sequence across trials of different conditions. Time-selective cells contributed at consistent times across all 9 target conditions (as described above). The contributions of row-selective and column-selective cells demonstrated structure that depended on the row or column of the target. To determine whether this was the case for a particular cell, the effect size was computed between the distributions of its spike times for each pair of conditions. The maximum effect size across condition pairs was considered. (For example, a cell with large effect size between the front and back rows could be considered row-selective, even if spike times for the front and middle rows were similar. The spike-times of such a cell would distinguish between the front and back row of a target, but make less distinction between the front 2 rows.) These effect sizes define a continuous measure of selectivity across cells. Cells with selectivity above a threshold percentile were labeled selective. Thresholds were determined separately for each subject, so that the population of selective cells maximizes

decoding performance of the condition for which they are selective (see Supplementary Fig. S5d–g for this procedure).

## Statistics and reproducibility

A complete statistics table with information about tests for all figures can be found in the supplement. Two male macaque monkeys participated in the experiments. There are no reported effects of sex in the variables of interest.

## Reporting summary

Further information on research design is available in the Nature Portfolio Reporting Summary linked to this article.

## Data availability

The sequences and processed data used for the analyses in this study can be found at this study's GitHub repository (https://github.com/mullerlab/buschEAsequences). The raw neural data used for this study are provided by Julio Martinez-Trujillo and are available upon reasonable request. Source data are provided with this paper.

## Code availability

All code associated with this study is available at https://github.com/mullerlab/buschEAsequences.

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

## Acknowledgements

This work was supported by the Canadian Institute for Health Research (CIHR) (J.M.), NSF/CIHR (NeuroNex Grant No. 2015276) (L.M., J.M.T.), NSERC (L.M., J.M.T.), Research in Autism from the Government of Ontario (J.M.T.), the Canadian Foundation of Innovation (CFI) (J.M.T), BrainsCAN at Western University through the Canada First Research Excellence Fund (CFREF) (L.M.), Compute Ontario (computeontario.ca) (L.M.), Digital Research Alliance of Canada (https://alliancecan.ca/en) (L.M.), SPIRITS 2020 of Kyoto University (L.M.), and the Western Academy for Advanced Research (L.M.). M.R. gratefully acknowledges support from the NSERC PGS-D. A.B. gratefully acknowledges support from the NSERC PGS-D and BrainsCAN Graduate Fellowship Program.

## Author contributions

Conceptualization: A.B., M.R., L.M., J.M.T.; data curation: A.B., M.R.; formal analysis: A.B., M.R., M.M.; funding acquisition: L.M., J.M.T.; experimental design: M.R., J.M.T., L.P., R.L., A.S., M.L.; development of virtual environment: M.R., R.G., B.C.; neural recordings: M.R., R.L., M.L.; investigation: A.B., M.R., L.M., J.M.T.; computational methods: A.B., M.R., J.M., L.M.; writing—original draft: A.B., M.R.; writing—review and editing: A.B., M.R., L.M., J.M.T.

## Competing interests

The authors declare no competing interests.
