## [Peer Review File · Nature Communications]

Neuronal activation sequences in lateral prefrontal cortex
encode visuospatial working memory during virtual navigationREVIEWER COMMENTS

Reviewer #1 (Remarks to the Author):

This is an important study, employing a paradigm typically used to examine memory in rodent hippocampus, but instead applied to the primate dorsolateral prefrontal cortex (dlPFC). The results are exciting and fruitful, helping to illuminate the similarities and differences in coding between these neural systems that have heretofore been studied in isolation, without the essential bridges needed for direct comparisons. The finding that the primate dlPFC is able to represent a sequence of future actions in working memory, and that this depends on NMDAR mechanisms, should be of great interest to both the prefrontal and hippocampal fields, and may help to bridge these disparate communities.

A few suggestions for improvement of the current paper:

1. The authors posit that this form of representation differs from the classic persistent firing model, but one could argue that instead it is actually an elaboration of this operation. Thus, might their data actually be the sequential activation of persistent firing (for about 250msec) of neurons/microcolumns that represent that particular location in visual space? Eg a “bump” moving through mnemonic space to represent a sequence in imagination/memory/planning space, where each microcircuit is only briefly activated to represent the rapid movement through space?

E.g. the brief and sequential activation of the recurrent excitatory microcircuits that are tuned for representing:

40°

44°

48°

52°

56°

60°

to hold in mind the progression of future events in virtual space? As the persistent firing hypothesis is related to the memory of a single location, it is logical that a sequence of locations would require multiple neurons/microcircuits devoted to differing locations to create the sequence in working memory. Thus, these working models are not in opposition, but rather, on deeper consideration, one may be the underlying component parts of the pattern seen here.

2. The authors describe the disruption of the pattern of representations with ketamine administration as due to ketamine altering the firing of interneurons, but it may also involve disruptions in the recurrent excitatory connections that depend on NMDAR and are reduced by ketamine. The authors analyzed the disruption in the representation of the sequences, but did not describe whether individual neurons showed increased or decreased firing, and whether these were regular- or fast-spiking neurons, which would help relate their physiological data to hypotheses regarding drug effects on dlPFC microcircuitry. As the primary location of NMDAR in primate dlPFC is on spines, and as persistent firing likely contributes to the “bursts” of activity sustained for 250msec to represent a location in mnemonic space, this mechanism needs to be added to their interpretation.

3. Small point:

Add “sequences” to the abstract:

Our results reveal a novel mechanism, NMDA-dependent neural activation sequences, for encoding sequences in visuospatial working memory in complex naturalistic environments. They further reveal the versatility and adaptability of neural codes supporting working memory function in the primate lateral prefrontal cortex.

Reviewer #2 (Remarks to the Author):

In the manuscript “Neuronal activation sequences in lateral prefrontal cortex encode visuospatial working memory during virtual navigation” by Busch and colleagues, the authors show that visuospatial working memory is encoded in the activation sequences of PFC neurons during a naturalistic working memory task.

The task and neural data provide a unique naturalistic perspective on working memory encoding. The conclusions are surprising and potentially transformative (if proven true). Assuming that the major comments are addressed, I think this manuscript would become a very valuable contribution to the literature.

Major Comments

1- Fig. S11. Surprisingly, this figure is relegated to the supplement. I say this is surprising since arguably the main contribution of this manuscript is that sequence coding explains the data better than the more conventional firing rate coding over time.

I do have some concerns about this analysis though:

- Are the number of cells matched across the decoders? From the description in Suppl L223 it does not seem so, which could induce biases (the decoder with more neurons included would perform better). I would suggest re-running this analysis matching the number of units across decoders (perhaps creating a distribution of performance for the subsampled decoder to determine whether the sequence decoder is significantly better).

- Furthermore, the power of employing machine learning techniques in neurobiological data is that this frees us from the burden of defining selective cells since all neurons can be included (and likely contain information, even if it doesn't survive the arbitrary selectivity criterion of researchers). I would suggest including an additional analysis that includes all cells regardless of selectivity (matching the numbers of cells included).

2- The comparison between sustained activity and sequence encoding rests on the assumption that non-peak periods do not contain information about the memoranda. Why not test this explicitly by excising the peak activity period of each cell in each trial and attempting to decode working memory information? I would bet that information would still be decodable, thus implying that sustained activity is either exclusively, or in addition to sequence coding, encoding working memory information.

Minor Comments

P8: "We did not find neural boundaries parcellating the epochs in this task (Fig. S6a)."

I am not sure this conclusion is warranted since NHP B shows a clear 4000ms peak (similar to the memory task). The difference is that NHP B does not show the pre-delay peak, and NHP T has no clear peaks during this period but has peaks at the start of the trial.

Relatedly,

P9: "trial parcellation by time boundary cells signalling the beginning of the working memory period was specific to the working memory task,"

This is a strange statement to make. The control task had no 'beginning of the working memory period' since there is no working memory period. How could a boundary cell signal that if it's absent? The only similar task transition between the two tasks is the point in which the monkey can start moving... and, at least in NHP B, there are clear 'time boundary' cells.

Are there neurons that contribute separately to the NAS of different periods? (i.e. mixed selective)

Fig. 4f. It does not strike me as a fair comparison to compare the correlation of a behavioral matrix that has clear clusters (navigation) with one that doesn't (ODR).

Figure 4f. Are comparisons between memory periods carried out only on the memory periods? Or also include navigation? If navigation is included, it would be useful to analyze this only during the delay period.

P13: "These results indicate a causal link between NMDA receptor dysfunction caused by ketamine and disruption of NASs leading to deficits in working memory."

What is the relationship between loss in elevated activity and NAS? Could NMDA effect be explained by the disruption of elevated activity rather than the disruption of NASs?

P15: "via NASs when the memoranda spatiotemporal structure varies, as we show here."

Why is it claimed that the memoranda varies? Isn't the location of the target (which is fixed in space) the memoranda? What does change is the position of the animal with respect to the memoranda, but this would not fit well with the argument raised in this paragraph.

P15: "the generation of working memory-related NASs relies specifically on NMDA receptor function."

Why 'specifically'? Is this meant to imply that it does not rely on other neurotransmitters? If not, I'd suggest removing this word.

P15: "A sequence code may also provide working memory systems with the temporal resolution to encode rich spatiotemporal information, such as all the relevant sensory and cognitive information involved in navigating to a target in the virtual reality task."

I am not sure what the authors tried to claim here. The memory information, which is what the sequence code is presumably encoding, doesn't contain 'rich spatiotemporal information'. Nowhere was it shown that the PFC contains information about sensory or cognitive processes other than the memory of the target location.

The sequence code has lots of issues, including the problem of what happens when the length of the delay varies... does the PFC recruit new cells to tile the additional delay? Does it cycle back to the original causing a redundant cycling code? Further, how is this code interpreted by downstream networks? I think it would be useful to discuss these and other drawbacks of this coding scheme.

P16: "Increased temporal resolution may also allow neurons to multiplex during different task periods, allowing for higher dimensional representations and flexible cognition."

I am not sure what the authors mean here when they mention 'temporal resolution'. Do they mean that the sequence code has more information than the rate code about the time elapsed during the memory delay period? If this is the claim, this should be shown explicitly (I did not see an analysis addressing this point).

Also, how does sequence code allow multiplexing? Are they referring to mechanisms such as the activity silent models of Mongillo, or Miller? If so, these models do not discuss sequence codes, since in those models activity peaks recur during the delay period.

I'm also unclear about how the sequence code may allow 'higher dimensional representations'. Higher than rate codes? Why so?

Movie S1 is referred to in the task description, but not references elsewhere or uploaded

Suppl, L17: "They are not required to stop at the target, but only a small radius of space will trigger a correct response and a juice reward"

Isn't this an issue with the task? The monkeys can employ a strategy where they just remember the column and navigate through each location in the column until a reward is reached.

Suppl L91: Should be S2g.

Suppl L131: "When the neurons were sorted in increasing order of t_n 131 values, clear bands of spike bursts are visible which span both arrays and recording length (Fig 1g,h)."

Yes, but you would also see these bands if you shuffled noise and sorted by peak activity, so I'm not sure that it is worthwhile emphasizing this as a meaningful result (or in the main text).

Why did the authors include in the memory code analysis the period of cue presentation and navigation? Wouldn't it be more meaningful (both for the nav task and the ODR task) to just use the delay period activity for the analyses?

Fig. S5C and suppl L137: "We used thresholds of 1s for NHP B and 2s for NHP T; however, we note that a 2D scan across thresholds (height and width) suggests the specific thresholds chosen does not significantly impact results."

How were the thresholds determined? And why are they different between monkeys? (1s vs 2s). If the threshold does not impact the results, why choose different ones?

Suppl L133: "Cells were considered time-selective if they contributed to the sequence at the same time, t_n , across trials of all 9 target conditions (i.e., their position in the sequence is consistent across trials and not related to target condition - see Fig. S5a,b for examples)."

I don't understand how this analysis ensures that the time-selective response occurs across all 9 target conditions. I initially thought that the authors had carried out a 2-way ANOVA (location/time) and identified neurons with time selectivity, but not location selectivity. But that is not what is done here.

Response to Reviewers - NCOMMS-23-49244-T

We thank the Reviewers for their careful consideration of our work, and the Editorial Team for the opportunity to submit a revised manuscript. We have carefully considered the Reviewers' comments on our manuscript, and specifically how to present our findings in relation to previous work on the persistent code for working memory. In the response below, and in our revised manuscript, we now specify in more detail (1) the specific model we consider for the persistent code, and (2) how it compares with the sequence code we present in this work.

The central point of our comparison is that the sequence code is distinct from the persistent activity code. Specifically, the sequence code is a temporal code, which relies solely on the timing of the cells' transient, elevated firing (without considering the rates within these elevations). On the other hand, the persistent code relies on the firing rates of a subpopulation of cells that exhibit persistent delay activity. Here, we define this to be activity that is (a) selective for individual stimuli throughout the delay period and (b) elevated compared to baseline for preferred stimuli.

It is important to emphasize that we are not making a comparison between the sequence code and firing rate codes in general, and we note further that the sequence code we report here can co-exist with firing rate codes (please see response to Reviewer 2, Point 2 for more details). Importantly, however, during the VR task, when we compare the sequence code to the persistent code defined above, the sequence code does better. During the ODR task, we find the reverse: the persistent code outperforms the sequence code.

Taking this together with the fact that the sequences are more correlated to behavior when the subject's visual perspective and navigation strategies are taken into account (as demonstrated in our original submission), we propose that the LPFC switches from a persistent firing code to a sequence code as task demands increase. Specifically, we propose that the LPFC uses a persistent code when tasks constrain eye position during the memory period and require stereotyped responses, such as saccades in the ODR task, but uses a sequence code when the task requires a spatiotemporally complex response, such as navigating through our VR environment, where eye position is unconstrained and therefore distractors and response strategies can be highly variable across trials (even of the same condition).

We appreciate that this central comparison is critical to make clearly if our work is to be understood by a general audience. In the revised manuscript, we have now worked to ensure that this comparison is stated very clearly. We believe that the revised manuscript, along with the point-by-point reply below, addresses all points raised in this round of review. We appreciate that the Reviewers' comments have led to a much stronger version of the manuscript, and hope that this response addresses all concerns.

We have included a point by point response to all comments below, with Reviewer comments in **black** text and our responses in blue. We have also attached an updated manuscript, with changes tracked in blue.

REVIEWER COMMENTS

Reviewer #1 (Remarks to the Author):

This is an important study, employing a paradigm typically used to examine memory in rodent hippocampus, but instead applied to the primate dorsolateral prefrontal cortex (dlPFC). The results are exciting and fruitful, helping to illuminate the similarities and differences in coding between these neural systems that have heretofore been studied in isolation, without the essential bridges needed for direct comparisons. The finding that the primate dlPFC is able to represent a sequence of future actions in working memory, and that this depends on NMDAR mechanisms, should be of great interest to both the prefrontal and hippocampal fields, and may help to bridge these disparate communities.

We thank the Reviewer for their positive assessment of our work. There are indeed some similarities between our task and memory navigation tasks used in hippocampus research. However, there are some important differences between our findings and sequences of place cells' activation (e.g., during replay), which we will discuss in the points below.

A few suggestions for improvement of the current paper:

1. The authors posit that this form of representation differs from the classic persistent firing model, but one could argue that instead it is actually an elaboration of this operation. Thus, might their data actually be the sequential activation of persistent firing (for about 250msec) of neurons/microcolumns that represent that particular location in visual space? Eg a "bump" moving through mnemonic space to represent a sequence in imagination/memory/planning space, where each microcircuit is only briefly activated to represent the rapid movement through space?

E.g. the brief and sequential activation of the recurrent excitatory microcircuits that are tuned for representing:

40° 44° 48° 52° 56° 60°

to hold in mind the progression of future events in virtual space? As the persistent firing hypothesis is related to the memory of a single location, it is logical that a sequence of locations would require multiple neurons/microcircuits devoted to differing locations to create the sequence in working memory. Thus, these working models are not in opposition, but rather, on deeper consideration, one may be the underlying component parts of the pattern seen here.

We thank the Reviewer for their interesting suggestion. To test this possibility, one must first define tuning for the cells in the delay sequences. Since the NHP is waiting at the start location and not moving during the delay period, we turn to the activity during the navigation period to define tuning preferences for each cell.

One way this tuning could occur is if the "bumps" that make up the navigation sequences occur at specific positions along a trajectory through virtual or visual space, and that these sequences of positions are replayed or rehearsed during the delay sequences. However, this is unlikely since the delay and navigation sequences are made up of disjoint sets of cells..

To further investigate the reviewer’s suggestion, we conducted an additional analysis. We consider that tuning during navigation could occur at different time scales and not necessarily associated to the transient activity that makes up the sequences. (i.e. during non-peak / non-sequence activity). If this were the case, sequences during the delay period could reflect the sequential activation of neurons that display such tuning for specific locations during navigation (Fig. R1a).

Figure R1 (a) Schematic: Delay sequences represent the sequential activation of neurons tuned for particular locations during navigation. (b) Firing fields of two example cells during navigation. Locations in the arena are binned into a ‘checkerboard’ and colored by the number of spikes per second of occupancy time in that bin. (c) Schematic of the Chebychev or “king’s chessboard” distance metric used in the following panels. If sequence activations trace out a trajectory through mnemonic space, the Chebychev distance between consecutive cells should be 1. (d) The observed distances between preferred locations for consecutive cells in the delay sequence, pooled across all sessions. (e) The same measure repeated for cells that participate in the sequence 10 positions apart (red line) and 20 positions apart (yellow shaded area).

We began by determining preferred locations for each cell using the data from the navigation period. To do so, we binned the VR arena into a “checkerboard” (see Fig. R1b, grid lines) and determined whether cells fired preferentially when the animal was navigating through specific bins (see Fig R1b for example cells). We note that for this analysis we considered only the targets in the first row (the white circles in Fig. R1b). The next question, as pointed out by the Reviewer, is whether the delay sequences represent the sequential (in time) activation of cells with sequential (in space) preferred locations (Fig. R1a, schematic). If this were the case, we should see a relationship between the distance in position between two cells in the delay sequence and the distance between their preferred locations. That is, cells that contribute to the sequence consecutively should also have preferred locations that are close together (in order to trace out a trajectory). Here, we use the Chebychev (or “king’s chessboard”) distance to measure distance

between preferred locations (Fig R1c). We do not find evidence in support of this idea: cells that participate consecutively in the delay sequence do not appear to have adjacent preferred locations (Fig R1d). Further, the distribution of distances between preferred locations is no different for cells that are consecutive in the sequence compared to cells that contribute to the sequence 10 or 20 positions apart (Fig R1e).

For this reason, we believe it is unlikely the sequences are made up of brief consecutive activations of neurons tuned to represent specific locations along a navigation path, and are therefore fundamentally different from place cell codes (e.g., spatial replay) in the hippocampus. However, we cannot discount the possibility that the Reviewer's suggestion could be at play in a different task-related space -- for example, the sequences could exist in a more abstract representation of space. We believe that this is a really interesting topic that needs further exploration, but is beyond the scope of this work. We hope to investigate these interesting possibilities in future work.

Finally, in the course of carrying out these new analyses of the navigation data, we discovered a minor bug in our code that caused additional time to be included in some trials in Fig. 4d. We have now corrected this issue. The results remain qualitatively unchanged and statistical significance was not affected, but we feel it is important to make the Editor and Reviewers aware that this panel now displays qualitatively similar but quantitatively different results.

2. The authors describe the disruption of the pattern of representations with ketamine administration as due to ketamine altering the firing of interneurons, but it may also involve disruptions in the recurrent excitatory connections that depend on NMDAR and are reduced by ketamine. The authors analyzed the disruption in the representation of the sequences, but did not describe whether individual neurons showed increased or decreased firing, and whether these were regular- or fast-spiking neurons, which would help relate their physiological data to hypotheses regarding drug effects on dIPFC microcircuitry. As the primary location of NMDAR in primate dIPFC is on spines, and as persistent firing likely contributes to the "bursts" of activity sustained for 250msec to represent a location in mnemonic space, this mechanism needs to be added to their interpretation.

We thank the Reviewer for raising this point. A recent work from our group (Roussy, *Molecular Psychiatry*, 2021), studying the same task and NHPs, has addressed these points more fully than the present manuscript. We refer to it in our paper. There, we report that the working memory deficits induced by ketamine were accompanied both by decreased firing of narrow spiking inhibitory interneurons (to all memorized locations) and increased firing of excitatory cells to non-preferred locations. These findings align with a previously proposed mechanism for working memory disfunction - that reduced NMDAR conductance of interneurons leads to generalized disinhibition of pyramidal cells, which results in a loss of tuning (and therefore of spatial specificity of WM representations) (Lisman, *Nature Neuroscience*, 1998; Wang, *J. Neurosci.*, 1999; Driesen *Neuropsychopharmacology*, 2013). We agree that this mechanism is important for our interpretation of ketamine's effect on the microcircuitry, and we have clarified this point in the updated manuscript (see pg 17). We are currently working on a separate manuscript to analyze

how Ketamine affects firing patterns of neurons within the LPFC. It is not a simple answer, and we hope the reviewer understands it goes beyond the scope of the manuscript. Here we limit our conclusions to the fact that Ketamine alters the sequence code.

3. Small point:

Add “sequences” to the abstract:

Our results reveal a novel mechanism, NMDA-dependent neural activation sequences, for encoding sequences in visuospatial working memory in complex naturalistic environments. They further reveal the versatility and adaptability of neural codes supporting working memory function in the primate lateral prefrontal cortex.

We thank the Reviewer for their suggestion and have updated the abstract to read: “Our results reveal a novel mechanism, NMDA-dependent neural activation sequences, for encoding working memory in the context of a rich and continuous visuospatial task.”

Reviewer #2 (Remarks to the Author):

In the manuscript “Neuronal activation sequences in lateral prefrontal cortex encode visuospatial working memory during virtual navigation” by Busch and colleagues, the authors show that visuospatial working memory is encoded in the activation sequences of PFC neurons during a naturalistic working memory task.

The task and neural data provide a unique naturalistic perspective on working memory encoding. The conclusions are surprising and potentially transformative (if proven true). Assuming that the major comments are addressed, I think this manuscript would become a very valuable contribution to the literature.

We thank the Referee for their positive comments and appreciate that the suggestions below have led to a stronger comparison between the sequence and persistent codes in our task.

Major Comments

1- Fig. S11. Surprisingly, this figure is relegated to the supplement. I say this is surprising since arguably the main contribution of this manuscript is that sequence coding explains the data better than the more conventional firing rate coding over time.

I do have some concerns about this analysis though:

- Are the number of cells matched across the decoders? From the description in Suppl L223 it does not seem so, which could induce biases (the decoder with more neurons included would perform better). I would suggest re-running this analysis matching the number of units across decoders (perhaps creating a distribution of performance for the subsampled decoder to determine whether the sequence decoder is significantly better).

- Furthermore, the power of employing machine learning techniques in neurobiological data is that this frees us from the burden of defining selective cells since all neurons can be included (and likely contain information, even if it doesn't survive the arbitrary selectivity criterion of researchers). I would suggest including an additional analysis that includes all cells regardless of selectivity (matching the numbers of cells included).

We thank the Reviewer for the suggestions and appreciate the opportunity to expand on these points. Figure S11 was intended as a specific test of two competing hypotheses. The first, proposed in this work, is that sequential activity in LPFC encodes WM content (temporal code). The second hypothesis is that memory content is encoded by a subset of cells that exhibit persistent delay activity (persistent firing code). We note an important distinction between the persistent code and "firing rate coding". The latter is a general framework and is not necessarily tied to a specific profile of firing activity (e.g., persistent or sustained). Based on the Reviewer's suggestions, we improved our analysis and we have now moved it to Fig. 5 of the main text (also copied below).

We address the Reviewer's concerns in the new analyses presented below. The number of cells were previously not matched between decoders due to the nature of the codes in question: the persistent firing hypothesis relates to a specific subpopulation of cells, which we have tried to identify using standard criteria for persistent activity. Below, we repeat the analysis while progressively relaxing these criteria to include substantially more cells in the persistent activity decoder, until they are close to matched with the number of cells participating in the delay sequences. Specifically, cells were included if: (1) they are tuned throughout the delay period, with tuning defined by a p-value threshold from an ANOVA across targets and (2) they display an elevated firing rate compared to baseline for their preferred targets (note: difference from baseline is also defined by a p-value threshold from an ANOVA). We considered cells that displayed tuning for up to three target locations. We highlight that cells displaying a strict definition of persistent firing were extremely rare in our dataset, but with a sufficiently relaxed definition, we were able to find a subset of cells displaying the hallmarks of delay activity (as demonstrated by the SDFs plotted below, which are averaged across cells and trials). We repeated this analysis using the same criteria during an ODR task performed by the same NHPs that had the same number of target locations (9, also in a grid layout), and we find far more persistent cells in the ODR task than the VR task. We believe this is an important result and we thank the reviewer for raising this concern.

Figure R2: Below, as a function of the p-value threshold for determining tuning and elevated firing, we plot (a) the fraction of cells included in the decoder, (b) the decoding accuracy, and (c) the averaged response to preferred targets (black), the averaged response to non-preferred targets (red), and 20 example single cell mean responses to preferred targets (gray). We note that the firing rates below are normalized to baseline and that we switched from a multinomial regression model to a support vector machine (SVM) decoder because it produced better decoding accuracies for the persistent cells, particularly as the number of cells included increased. (d) Fraction of cells contributing to each decoder in the ODR task. (e) Decoding

accuracy for targets in the ODR task. (f) The averaged response to preferred targets in the ODR task (black), the averaged response to non-preferred targets (red).

In the VR task, the sequence decoder performs significantly better than all versions of the persistent decoder, even as the number of persistent cells is increased to match the fraction of cells in the delay sequences. In the ODR task, on the other hand, there are substantially more persistent cells, and the persistent decoders perform significantly better than the sequence decoder. This makes sense, considering we have shown that single cells peak activations are temporally inconsistent in the ODR task (Fig. S8d,e), resulting in unreliable sequences with a weak relationship to behavior (Fig. 4). Further, if we restrict the criteria for persistent activity so that the fraction of persistent cells in ODR matches previous reports (Luna et al., *Journal of Vision*, 2019), we find no persistent cells in the VR task.

Finally, we agree that definitions of selectivity are defined by the statistics researchers choose, and that applying machine learning techniques to the activity of the full population may reveal interesting trends. We presented decoding analyses of the same data used here in two previous papers (Roussy et al. 2021, 2022). Here we specifically used decoders to probe sequences and to probe persistent firing. We believe the analyses we just conducted show the utility and versatility of decoders for hypothesis testing.

2- The comparison between sustained activity and sequence encoding rests on the assumption that non-peak periods do not contain information about the memoranda. Why not test this explicitly by excising the peak activity period of each cell in each trial and attempting to decode working memory information? I would bet that information would still be decodable, thus implying that sustained activity is either exclusively, or in addition to sequence coding, encoding working memory information.

We appreciate the Reviewer's suggestion. We removed the peak elevations and attempted to decode the target from the remaining activity of the cells that contribute to the delay sequences. We find that decoding performance drops significantly when we do this (mean non-peak activity = 28%, mean sequences = 38%, chance ~ 11%, paired t-test, $p=1.16e-5$).

Minor Comments

P8: "We did not find neural boundaries parcellating the epochs in this task (Fig. S6a)."

I am not sure this conclusion is warranted since NHP B shows a clear 4000ms peak (similar to the memory task). The difference is that NHP B does not show the pre-delay peak, and NHP T has no clear peaks during this period but has peaks at the start of the trial.

Relatedly,

P9: "trial parcellation by time boundary cells signalling the beginning of the working memory period was specific to the working memory task,"

This is a strange statement to make. The control task had no 'beginning of the working memory period' since there is no working memory period. How could a boundary cell signal that if it's absent? The only similar task transition between the two tasks is the point in which the monkey can start moving... and, at least in NHP B, there are clear 'time boundary' cells.

We thank the Reviewer for raising this point. We did not discuss the peaks mentioned by the reviewer (P8) because we did not see consistent peak times across the two subjects in the control task (or after ketamine). We do, however, agree that the phrasing for the noted sentence on P9 required some edits. In the revised manuscript, this sentence now reads: "trial parcellation by time boundary cells (that signal the beginning and end of the working memory period) was specific to the working memory task because the offset of the cue was specific to the working memory task". Now it is clear we referred to the delay period and not to the entire trial. This result is interesting to us because the signal provided by time consistent cells is internally generated and varies between the tasks depending on the subject's expectation for that task (the WM task, which has an early cue offset defining the delay period and requires WM maintenance, versus the control task, which does not have the cue offset neither delay period and does not require WM). Our intention in making the statement is to indicate that the peak appears only when the task has a defining boundary (cue offset) signaling the beginning of the WM period, and does not appear in the absence of task boundaries. It is possible that boundary neurons may signal expectations for task events in general (not specifically WM), which could explain why we see boundary activity for NHP B when navigation starts in the perception-control task as the Reviewer notes. We clarify this now in the text.

Are there neurons that contribute separately to the NAS of different periods? (i.e. mixed selective)

This is an interesting suggestion. However, since the sequences are defined based on the peak firing time during the trial, each cell could only contribute to the NAS during a single trial period. We did find cells that contributed to the sequence during different trial periods for different targets (see for example Fig. S5d). However, if we allow cells to contribute separately to the sequences

during each trial epoch (“all cells”), the decoding performance decreases compared to allowing each cell to contribute only once during the full trial (“sparse delay sequences”).

Figure R3: The column decoding accuracy is plotted for (left) the task period delay sequences, defined as in the manuscript where a given neuron can contribute once to the sequence, and (right) a sequence consisting of all cells, each contributing at the time they reach their maximum firing rate during the delay period. Here one cell contributed to the sequence during each task period.

Fig. 4f. It does not strike me as a fair comparison to compare the correlation of a behavioral matrix that has clear clusters (navigation) with one that doesn't (ODR).

We appreciate the Reviewer's concern. However, this is one of our main points: we found that the clusters in the behavioral matrix based on VR trajectories corresponded to the clusters of sequences, and that the correspondence was higher (i.e., correlation) than when using Euclidean distances between target locations. There was no a priori reason to assume that the sequences would better correlate with one or the other measurement. However, we understand the Reviewer's concern about comparing these correlations to those in the ODR task, where the behavioural matrix does not have clusters.

To address this point, we repeated the analysis using the matrix of Euclidean distances between target locations in VR. This destroys the clusters that were present due to the columnar groupings of trajectories and provides a more direct comparison to the behavioral matrix in the ODR task (Euclidean distances between target locations on-screen). Even in this case, we see significantly higher correlations between the sequences in VR and behavior compared to the sequences in ODR. Please see Fig. R2 below.

Figure R4: (Left) Behavioral distance matrix for VR. Note the naming scheme: targets 1-3 represent the left column from front to back, targets 4-6 are the middle column from front to back, and targets 7-9 are the right column from front to back. (Right) Correlation analysis presented in Fig. 4 replicated with the non-clustered behavioral matrix on the left.

Figure 4f. Are comparisons between memory periods carried out only on the memory periods? Or also include navigation? If navigation is included, it would be useful to analyze this only during the delay period.

We appreciate the Reviewer raising this important point. Navigation was not included in this analysis, since there is no navigation in the ODR task. We have now noted more clearly in the figure's description that this comparison is for delay period sequences.

P13: "These results indicate a causal link between NMDA receptor dysfunction caused by ketamine and disruption of NASs leading to deficits in working memory."

What is the relationship between loss in elevated activity and NAS? Could NMDA effect be explained by the disruption of elevated activity rather than the disruption of NASs?

We thank the Reviewer for raising this point. A recent work from our group (Roussy, *Molecular Psychiatry*, 2021), studying the same task and NHPs, reported that the working memory deficits induced by ketamine were accompanied both by decreased firing of narrow spiking putative inhibitory interneurons to all remembered locations and increased firing of broad spiking putative excitatory cells to the non-preferred remembered locations. Both types of cells lost tuning as a result. These findings align with a previously proposed mechanism for working memory dysfunction - that at low doses of Ketamine reduced NMDAR conductance of interneurons leads to generalized disinhibition of pyramidal cells, which results in a loss of tuning (and therefore of spatial specificity of WM representations) (Lisman, *Nature Neuroscience*, 1998; Wang, *J. Neurosci.*, 1999; Driesen *Neuropsychopharmacology*, 2013). A possible, speculative mechanism for sequence disruption is that the decreased firing of PV interneurons and resulting general increase in pyramidal cell firing causes the brief peaks that form the sequences to be less specific. This would decrease the time consistency of single cell contributions to the sequences and disrupt the sequence code. In this manuscript, we report the disruption of NASs by low doses of ketamine. Unfortunately we did not record from a large population of narrow spiking interneurons and our sample size did not allow investigating the exact mechanism of the effect.

P15: “via NASs when the memoranda spatiotemporal structure varies, as we show here.”

Why is it claimed that the memoranda varies? Isn't the location of the target (which is fixed in space) the memoranda? What does change is the position of the animal with respect to the memoranda, but this would not fit well with the argument raised in this paragraph.

This is an important point. We agree: the memoranda (target location in the VR) does not vary over time in 'spatial coordinates' - we have over simplified our point in the above statement. The point we were trying to make was that in the VR task eye position, distractors, and response paths can vary from trial to trial (even across trials of the same condition). Changes in eye position produce changes in the retinal position of the target bringing spatiotemporal complexity to the task. On the other hand, in the ODR task, the target location remains fixed in both retinal and space coordinates and responses are saccades, which are more stereotyped and ballistic. The ODR task is less spatiotemporally complex. We find that the persistent code outperforms the NAS code in the ODR task, but that the NAS code takes over in the VR task.

We have updated this sentence to read: “via NASs when the task requires responses with more complex spatiotemporal structure, as we show here,” and have made sure to more clearly define the boundary between results and interpretations (in regard to the Reviewer's comment 2 points below).

P15: “the generation of working memory-related NASs relies specifically on NMDA receptor function.”

Why 'specifically'? Is this meant to imply that it does not rely on other neurotransmitters? If not, I'd suggest removing this word.

We appreciate the suggestion and have updated the text accordingly: “the generation of working memory related NASs seems to depend on NMDA receptor function, though other neurotransmitter systems may also be involved.”

P15: “A sequence code may also provide working memory systems with the temporal resolution to encode rich spatiotemporal information, such as all the relevant sensory and cognitive information involved in navigating to a target in the virtual reality task.”

I am not sure what the authors tried to claim here. The memory information, which is what the sequence code is presumably encoding, doesn't contain 'rich spatiotemporal information'. Nowhere was it shown that the PFC contains information about sensory or cognitive processes other than the memory of the target location.

The reviewer raises an important point. We demonstrate that the sequences encode the memoranda in the task (target position in the VR environment), and in this sentence we speculate about why a sequence code may be used in our VR task.

The varying spatiotemporal information we are referring to here is the environment containing virtual 3D elements, changes in the scene caused by eye movements (which cause changes in

the target position in retinal coordinates), and the response of the animals which is spatiotemporally complex (a 3D path toward the target location that accounts for view in the virtual environment) compared to classic working memory tasks. In tasks such as ODR, the memoranda are stationary in a 2D screen and, when controlling for eye movements during the delay period, they are also stationary on retinal coordinates. Our point is that our task adds more complex dynamics than previous tasks exploring working memory correlates and that NAS are suited for encoding such dynamics.

That said, the point of the reviewer is well taken. Our results are a first step in investigating the NAS code and we need more experiments to characterize how complex varying spatiotemporal information is encoded by LPFC neurons and ensembles. We have revised the sentence to read: "We speculate that a sequence code may also provide working memory systems with the temporal resolution to encode varying spatiotemporal information, such as changes in 3D scenery involved in navigating to a target in the virtual reality task or in the real world. This hypothesis, however, requires further testing."

The sequence code has lots of issues, including the problem of what happens when the length of the delay varies... does the PFC recruit new cells to tile the additional delay? Does it cycle back to the original causing a redundant cycling code? Further, how is this code interpreted by downstream networks? I think it would be useful to discuss these and other drawbacks of this coding scheme.

The Reviewer raises important points. These are all interesting questions. Unfortunately, we don't have all the answers yet, but as the reviewer suggests, we can add some lines to the discussion. Since we are able to record from only a very small subsample of cells in this area, we hypothesize that many other cells could be participating in population patterns like NASs. As such, rather than a cycling code, we anticipate additional cells would be recruited, or perhaps the timing of the sequences would adapt to changing delay lengths. Interestingly, hippocampal sequences that replay the path taken by a rat to a location, can undergo time compression (Nadasdy et al. 1999). Our experimental design may not be optimal to test this question, but this is something we plan to do in the future. We hope the reviewer understands this goes beyond the scope of the study.

Previous computational work has clearly demonstrated how sequences can be decoded by downstream neurons (Masquelier & Thorpe, *PLoS ONE*, 2008). However, we agree that there are open questions about codes like these, and have updated the discussion accordingly. (See the last paragraph of the discussion, page 17).

P16: "Increased temporal resolution may also allow neurons to multiplex during different task periods, allowing for higher dimensional representations and flexible cognition."

I am not sure what the authors mean here when they mention 'temporal resolution'. Do they mean that the sequence code has more information than the rate code about the time elapsed during the memory delay period? If this is the claim, this should be shown explicitly (I did not see an analysis addressing this point).

Also, how does sequence code allow multiplexing? Are they referring to mechanisms such as the activity silent models of Mongillo, or Miller? If so, these models do not discuss sequence codes, since in those models activity peaks recur during the delay period.

I'm also unclear about how the sequence code may allow 'higher dimensional representations'. Higher than rate codes? Why so?

We appreciate the opportunity to clarify this point.

By "increased temporal resolution", we were referring to the temporal scale of the code. In the schematic below, the vertical bars denote "time bins" that define a neuron's contribution to the code. A neuron exhibiting persistent firing (left) contributes throughout the delay period, whereas a neuron participating in a sequence (middle) contributes only for the short period of peak activity. As such, the "time bins" are shorter during the sequence, resulting in increased temporal resolution. We did not intend to claim that the sequences represent time elapsed during the trial. This would require additional analyses and experimental manipulations we did not to - for example, varying the length of the delay period - and would need to be addressed in future studies.

By "multiplexing", we were not referring to activity silent models, but rather to the increased flexibility of temporal codes in general (Mainen and Sejnowski, *Science*, 1995), and how they can allow multiplexing of different spike patterns (Krüger and Becker, *TINS*, 1991; McClurkin et al., *Science*, 1991). For example, if we consider a population of neurons displaying persistent activity (Fig. R5, left), the relevant time scale could be considered to be the entire epoch of sustained activity, until the response ("bin size", left panel, Fig. R5). With a population exhibiting a sequence code, on the other hand, it is possible that one sequence could occur within the population (middle panel, Fig. R5), or that several sequences could be superimposed in time (right panel, Fig. R5). With this in mind, we speculate that higher dimensional representations could be possible with the increased temporal resolution of a sequence code. This increased flexibility could, further, explain why we observe the sequence code being more prominent in the VR task than in ODR (cf. new Fig. 5). However, we have removed this sentence from the discussion to avoid further speculating about what can be a complex issue.

Figure R5 Schematic comparing the temporal resolution of codes relying on persistent, sustained activity versus sequential activity.

Movie S1 is referred to in the task description, but not references elsewhere or uploaded

We thank the Reviewer for noting this point. This was intended to be a link to a video of an example task trial. In the updated submission, we have corrected this issue.

Suppl, L17: “They are not required to stop at the target, but only a small radius of space will trigger a correct response and a juice reward”

Isn't this an issue with the task? The monkeys can employ a strategy where they just remember the column and navigate through each location in the column until a reward is reached.

We appreciate the Reviewer's point, and agree that the task design is likely related to the column clustering we see. However, we note that if this were the strategy, we would see only stereotyped trajectories leading straight down the columns. Instead, the subjects took curved trajectories that often followed the edges of the arena (outside the reward zones for other targets), and that changed from trial to trial. We have included some additional examples of trajectories below, that demonstrate multiple strategies to reach the same target, such as going down a different column and then doubling back or turning to one side. This is an interesting phenomenon, which did not happen on the majority of trials, and we would like to study it further. For now, we take it as evidence that the animals navigated towards a particular target location.

Figure R6: Example correct trajectories for 2 recording sessions. Example trajectories are highlighted in red and blue that arrive at the same target by following different paths.

Suppl L91: Should be S2g.

This has been updated, thank you.

Suppl L131: “When the neurons were sorted in increasing order of t_n 131 values, clear bands of spike bursts are visible which span both arrays and recording length (Fig 1g,h).”

Yes, but you would also see these bands if you shuffled noise and sorted by peak activity, so I'm not sure that it is worthwhile emphasizing this as a meaningful result (or in the main text).

We appreciate the opportunity to expand on this point. Two main qualities of the sequences were surprising to us. First is the fact that we see these brief elevations in activity at all, and that they

stand out so far above the background firing. These peaks are not found in shuffled data (see Fig. S3a, copied below). Second, these peak responses tile the SDF plots in a specific way -- spanning both the full array and the full length of the trial. There are many possible sequences that could be obtained, particularly by sorting as the Reviewer suggests: for example, sequences that include all cells but are much shorter in time (resulting in a much steeper slope), or sequences that include only a subset of cells. When we initially discovered the sequences in our data, this particular tiling led us to consider that the sequences may be related to the task. Finally, even if bands of activity like these could be obtained by shuffling noise and sorting by peak activity, the ordering of these bands would be random on each trial and could not display any task-related structure such as the sequences we report here. Also, observe that in the examples shown in Fig. 2c,d, the activation of individual neurons occurs at the same time (x axis) in different trials (y axis). We have now addressed this point more clearly in the text. (See P5 in the main text and L132 in the supplement).

Figure R7: To produce the plot to the left, we aligned the responses of the cells such that their peak firing rate occurs at $x=0$. We then normalized the responses and averaged across cells and trials (blue line). We repeated the same procedure after shuffling the spikes of each cell within each trial (grey line) to confirm the peak firing responses that make up the sequences in our data do not occur by chance.

Why did the authors include in the memory code analysis the period of cue presentation and navigation? Wouldn't it be more meaningful (both for the nav task and the ODR task) to just use the delay period activity for the analyses?

We appreciate the Reviewer's suggestion. We analyzed each period separately in the VR task because we were interested in the relationship between sequences across trials, and curious if they were specific to the delay period. In the VR task (Fig. 3), we considered the sequences defined by neural boundaries (which includes some of the cue period, and does not include the complete delay period). We did, however, conduct the same analysis for the delay period (defined by the task boundaries), which can be found in Fig. S4c. Further, the analysis comparing VR and ODR only considers the delay period sequences (as the Reviewer suggests) in order to match the time window between both tasks.

Fig. S5C and suppl L137: "We used thresholds of 1s for NHP B and 2s for NHP T; however, we note that a 2D scan across thresholds (height and width) suggests the specific thresholds chosen does not significantly impact results."

How were the thresholds determined? And why are they different between monkeys? (1s vs 2s). If the threshold does not impact the results, why choose different ones?

We thank the Reviewer for raising this point. We used a wider threshold for NHP T in order to include a comparable fraction of cells as NHP B. NHP T had fewer well isolated units, and slightly reduced temporal consistency overall, so for consistency of peaks across all recording sessions with NHP T, we widened the threshold. In Figure R5 below, we show that if we use, for example, a threshold of 1.5s for both subjects, the results are qualitatively unchanged (with peaks in the same location). We have noted this more clearly in the updated manuscript.

Figure R8: Times at which time-selective cells contribute to the sequence, using the same threshold across subjects.

Suppl L133: “Cells were considered time-selective if they contributed to the sequence at the same time, t_n , across trials of all 9 target conditions (i.e., their position in the sequence is consistent across trials and not related to target condition - see Fig. S5a,b for examples).”

I don't understand how this analysis ensures that the time-selective response occurs across all 9 target conditions. I initially thought that the authors had carried out a 2-way ANOVA (location/time) and identified neurons with time selectivity, but not location selectivity. But that is not what is done here.

We appreciate the Reviewer's point. The criteria we used for time-selectivity required the cell to achieve its peak firing rate at the same time (within a threshold as discussed above) across all trials of all nine conditions. Since the timing of the cell's peak response did not depend on the target location, and was consistent across all trials, we considered these to be time-selective cells. It is possible that these cells could display selectivity for target location in the rate of their response (as determined by an ANOVA), or in some non-peak activity, but since our analyses rely only on the sequence timing, we did not take this into account.

REVIEWERS' COMMENTS

Reviewer #1 (Remarks to the Author):

My concerns have all been very thoughtfully addressed

Reviewer #1 (Remarks on code availability):

NA

Reviewer #2 (Remarks to the Author):

I appreciate the thoroughness of the authors' responses. They have addressed all the questions raised in a very clear manner. The only point I was not very clear about was the discussion of the tiling of responses in Figure R7.

Figure R7 shows that peaks in the data are higher than the peaks that you would expect by shuffling the spikes trial-by-trial. This is not surprising, since we know that the task evokes changes in activity at specific times (this is true for any task. Even the ODR task).

This does not, however, test the hypothesis that the tiling observed would not be observed in data shuffled after the average is taken (not point by point and trial by trial, but by randomly displacing the averaged time series).

Reviewer #2 (Remarks to the Author):

I appreciate the thoroughness of the authors' responses. They have addressed all the questions raised in a very clear manner. The only point I was not very clear about was the discussion of the tiling of responses in Figure R7.

Figure R7 shows that peaks in the data are higher than the peaks that you would expect by shuffling the spikes trial-by-trial. This is not surprising, since we know that the task evokes changes in activity at specific times (this is true for any task. Even the ODR task).

This does not, however, test the hypothesis that the tiling observed would not be observed in data shuffled after the average is taken (not point by point and trial by trial, but by randomly displacing the averaged time series).

The reviewer is correct that transient elevations can also be found in shuffled data. What may be noticeable in Figure R7 is the magnitude of the peaks. We note that the focus of this point was to describe the basic features of the sequences that we observe relative to shuffled data. Although in the ODR task, we do see peak activations that can be organized within a trial in a sequential manner, there is no consistency in the tiling of peak activation times for a given target location (Fig. S8c). This contrasts with the specific tiling of peaks we observe in the VR task, where the order of the sequences is unique for each target and can be used to decode WM content on individual trials. Because this point was not critical to the findings, we have now removed this sentence from the paragraph on page 4. We believe the revised text better highlights the most interesting features of the sequences — that the ordering of the sequences on single trials encodes WM content and is disrupted by low doses of ketamine.

We again thank the Reviewer for their effort in considering this work. Their suggestions in this and the previous round of review have led to an improved version of the manuscript, and we hope this update addresses this last remaining point.

Summary:

The brain can maintain and flexibly manipulate complex visuospatial information 'in mind', an ability known as working memory. The neural codes underlying this function have been a matter of debate. This team simultaneously recorded the activity of hundreds of neurons in the lateral prefrontal cortex of male macaque monkeys during a visuospatial working memory task that required navigation in a virtual 3D environment. During task trials, animals perceived the location of a transient visual cue, remembered that location for a few seconds, and finally navigated toward it using a joystick to collect a reward. Authors report time boundary neurons that transiently activated just before the beginning and end of the memory

period. Moreover, distinct neuronal activation sequences (NAS) encoded specific remembered target locations in the virtual 3D environment, as viewed from the subject's own visual perspective. This NAS code outperformed the persistent firing code for the remembered location during the virtual reality task. However, during a classical working memory task using stationary stimuli and constraining eye movements, the persistent firing code outperformed the NAS code. Finally, blocking NMDA receptors using low doses of ketamine selectively deteriorated the NAS code during the working memory task. These results reveal a mechanism, NMDA-dependent NAS, for encoding working memory during a virtual reality task that resembles naturalistic settings. They further reveal the versatility and adaptability of neural codes supporting working memory function in the primate lateral prefrontal cortex.